# EXPLORING MEMORIZATION IN FINE-TUNED LANGUAGE MODELS

## ABSTRACT

LLMs have shown great capabilities in various tasks but also exhibited memorization of training data, thus raising tremendous privacy and copyright concerns. While prior work has studied memorization during pre-training, the exploration of memorization during fine-tuning is rather limited. Compared with pre-training, fine-tuning typically involves sensitive data and diverse objectives, thus may bring unique memorization behaviors and distinct privacy risks. In this work, we conduct the first comprehensive analysis to explore LMs' memorization during fine-tuning across tasks. Our studies with open-sourced and our own fine-tuned LMs across various tasks indicate that fine-tuned memorization presents a strong disparity among tasks. We provide an understanding of this task disparity via sparse coding theory and unveil a strong correlation between memorization and attention score distribution. By investigating its memorization behavior, multi-task fine-tuning paves a potential strategy to mitigate fine-tuned memorization.

## 1 INTRODUCTION

Large language models have demonstrated impressive capabilities in natural language understanding and generation, enabling significant advances across diverse applications including reading comprehension, text classification, and summarization (OpenAI, 2023; Ouyang et al., 2022; Bai et al., 2022; Touvron et al., 2023b). However, recent works reveal that pre-trained LMs tend to memorize and regenerate segments of their pre-training corpus when prompted appropriately. For example, Carlini et al. (2021) devised a training data extraction attack, successfully extracting hundreds of verbatim text sequences from GPT-2's training data. Various factors that affect memorization have been investigated and it is evident that memorization effects grow with model scale, data duplication, and prompt length (Lee et al., 2021; Kandpal et al., 2022; Carlini et al., 2022). This raises privacy and confidentiality concerns, as interactions with deployed LMs could enable extraction of the memorized sensitive training data, such as phone numbers, people's names, etc. As the scale of LMs and their training data continues to expand, the privacy risks posed by memorization become increasingly serious.

In addition to pre-training, the application of LMs often involves fine-tuning on downstream tasks (Touvron et al., 2023b; Chung et al., 2022; Ouyang et al., 2022; Longpre et al., 2023). Compared to pre-training, fine-tuning introduces two unique perspectives with respect to memorization. **First, fine-tuning often utilizes domain-specific, private, or valuable data.** For instance, developing a diagnostic chatbot (Yunxiang et al., 2023) requires collecting sensitive medical conversation data. Similarly, an academic LM (Beltagy et al., 2019) may be trained on copyrighted passages for summarization or paraphrase generation. Leakage of such fine-tuning data can seriously violate user privacy or intellectual property rights. **Second, fine-tuning involves more complex and diverse training objectives compared to pre-training.** Such differences may induce distinct memorization behaviors and patterns during fine-tuning. The aforementioned perspectives suggest that exploring memorization for fine-tuning is also necessary, but previous insights and findings regarding the pre-trained models may not apply to fine-tuning. Thus, dedicated efforts are desired to investigate LM's fine-tuning stage memorization.

To bridge this gap, we focus on the memorization of LMs during fine-tuning. We study various fine-tuning tasks including summarization, dialogue, question answering, machine translation, and sentiment analysis. Using an automatic plagiarism detection pipeline (Lee et al., 2023), we conduct studies on both popular open-sourced models, as well as the models that we fine-tuned for

diverse tasks. In both cases, we identify the existence of substantial memorization under certain tasks. Moreover, we discover potential factors that may impact memorization. Our key findings and contributions on memorization during LM fine-tuning are summarized as follows:

- *Fine-tuned Memorization presents a strong disparity among tasks.* For example, tasks like Summarization, and dialog present high memorization, while tasks like classification, reading comprehension, and translation show negligible memorization. This could be because different tasks require LMs to memorize different amounts of input features and we further justify this potential understanding based on sparse coding models.
- *Attention patterns are highly related to fine-tuned memorization.* For example, high memorization tasks demonstrate uniform, sparse attention distributions, contrasting with concentrated attention distribution for low memorization tasks. We provide an understanding on the relation between attention and memorization.
- *Multi-task fine-tuning can mitigate the memorization.* Multi-task fine-tuning can alleviate memorization compared to single-task fine-tuning.

## 2 RELATED WORK

Powered by the transformer architecture (Vaswani et al., 2017), in recent years, LMs such as Chat-GPT (Ouyang et al., 2022), Claude (Bai et al., 2022), Palm (Chowdhery et al., 2022), Llama(Touvron et al., 2023b;a) and T5(Raffel et al., 2020) have achieved impressive performance across a wide range of NLP tasks. The development of these large pre-trained models has been enabled by the increase in model size, data, and computing. These language models undergo pre-training by utilizing massive amounts of data to enhance their overall proficiency. Subsequently, people usually utilize various techniques (Chung et al., 2022; Ouyang et al., 2022; Houlsby et al., 2019; Hu et al., 2021; Li & Liang, 2021; Lester et al., 2021; Liu et al., 2021) to fine-tune the pre-trained models, thus enabling them to more effectively adapt to different downstream tasks.

The memorization behavior of pre-trained LMs has attracted increasing attention in recent years. Carlini et al. (2021) first proposed a training data extraction attack, demonstrating that LMs tend to memorize and regenerate segments of training data. Kandpal et al. (2022) and Lee et al. (2021) revealed that duplicated training data is more prone to memorization, and de-duplication can effectively reduce memorization. (Carlini et al., 2022) further quantified memorization effects, finding that memorization grows with model scale, data duplication, and prompt length.

There are also works providing different views and understandings on memorization. For example, Ippolito et al. (2022) developed an efficient defense preventing verbatim memorization, yet showed it fails to prevent leakage of training data. This shows the need for definitions beyond verbatim memorization. To distinguish "common" memorization from "rare" memorization, (Zhang et al., 2021) formulated a new notion of counterfactual memorization, which measures how a model's predictions change if a particular document is deleted during training. (Biderman et al., 2023) investigated predictable memorization by extrapolating small or partially-trained LMs' behavior to forecast memorization in larger models. They further presented scaling laws of prediction and gave recommendations to improve prediction reliability.

While most literature has focused on memorization during pre-training, limited work has investigated memorization arising in LMs' fine-tuning stage. (Mireshghallah et al., 2022) examined memorization risks in different fine-tuning methods for large LMs. They found that fine-tuning only the head leads to higher memorization compared to fine-tuning smaller adapter modules. (Lee et al., 2023) studied plagiarism during fine-tuning, concluding that fine-tuned LMs' plagiarism patterns depend on corpus similarity and homogeneity. However, these studies have considered fine-tuning with the same objective as pre-training, unlike most practical fine-tuning involving different tasks. In contrast to existing works, we focus on the more general and realistic scenario of multifaceted fine-tuning across diverse objectives.

## 3 PRELIMINARY

In this section, we first briefly introduce the definition and detection method of memorization in fine-tuned LMs, and then introduce preliminary findings on open-sourced fine-tuned LMs across various tasks.

## 3.1 DEFINITIONS AND NOTATIONS

**Memorization in Pre-training.** Given a language model $f(\cdot)$ which takes inputs from text $x$ and outputs the completion, previous works have various definitions of memorization in the pre-training stage. Previous work (Carlini et al., 2022) made a straightforward and strict definition that a string $s$ is extractable with its context $p$ (with length $k$) if the concatenation $[p\|s]$ exists in the training set and $f(p)$ produces exactly the same output of $s$, i.e., $f(p) = s$. Such a definition is called verbatim memorization. (Ippolito et al., 2022) also gives a relaxed definition of memorization called approximate memorization. They compare the similarity between $f(p)$ and $s$ using Bilingual Evaluation Understudy(BLEU) scores, which operate by comparing n-grams of two sentences. The sentence $s$ is identified as memorized if $\text{BLEU}(f(p), s) > 0.75$. Furthermore, some works such as (Lee et al., 2023) make use of plagiarism detection tools to compare the machine-generated text with the whole training set to identify memorization.

**Identify Memorization in Fine-tuning.** Note that in fine-tuning, the models are usually trained to fulfill certain capacities, such as sentiment analysis, dialogue, and summarization. Therefore, we define the fine-tuning process as a supervised-training manner, with training samples $\mathcal{D}_{\text{train}} = \{(x_i, y_i)\}_{i=1}^n$, where $y_i$ is the desired output text given the input text $x_i$. Since the input texts can contain more information in our considered tasks, we majorly discuss the potential information leakage from the input corpus in the training set, which is $\mathcal{D}_{\text{input}} = \{x_i\}_{i=1}^n$.

To explore memorization, we follow the prompting approach (Carlini et al., 2022) by dividing each $x_i = [p_i\|s_i]$ to a length-$k$ prefix $p_i$, and a suffix $s_i$. We further define the set of all prefixes in the training set as $P = \{p_i\}_{i=1}^n$, and the set of all suffixes as $S = \{s_i\}_{i=1}^n$. We study the possibility for an LM user to leverage certain algorithms (discussed below) to extract the information about any $s_j$ in $S$ based on any prefix $p_i \in P$. In detail, given a dataset of size $n$ with suffix space $S$, we employ the local search engine, Elasticsearch[1], to identify the top-$K$ corpus candidates $S_K^i = \{s_1^i, s_2^i, \ldots, s_K^i\}$ that exhibit similarities to $f(p_i)$.

In our paper, we claim a successful case of memorization identification, by inputting the model output $f(p_i)$ and $s_j$ to a publicly available plagiarism detection tool $D$ to see if $D(f(p_i), s_j) = \text{True}$. In detail, we utilize the PAN2014 plagiarism detection tool[2] to assess the similarity between $f(p_i)$ and each candidate $s_j \in S_k^i$. This tool is capable of detecting the presence of plagiarised word piece pairs $(d_i, d_j)$, where $d_i$ and $d_j$ are word pieces from $f(p_i)$ and $s_j$, respectively. Once the plagiarism is confirmed, we identify it as a memorization case. We then count the number of all cases and divide by $n$ to get the memory ratio. This ratio serves as an indicator of the memorization behavior exhibited by the model. Furthermore, these memorized pieces are classified into three distinct categories of memorization, we also report them in our results, and include typical cases in Appendix E:

- **Verbatim**: $d_j$ is an exact replica of $d_i$.
- **Paraphrase**[3]: $d_j$ is a rephrased version of $d_i$ (synonymous substitution, word reordering, or back translation)
- **Idea Memorization**: $d_j$ encapsulates the fundamental essence of $d_i$ but is presented in a more comprehensive manner.

More implementation details and descriptions of these tools are included in the Appendix A.

## 3.2 PRELIMINARY FINDINGS

To initially explore the memorization effects during fine-tuning, we examined several popular open-sourced fine-tuned models from HuggingFace[4] that were fine-tuned on 6 representative tasks. These tasks include summarization, medical dialog, question&answering (QA), translation, and sentiment analysis. The results are shown in Table 1. The preliminary results in Table 1 suggest that substantial memorization of the fine-tuning data could occur at the fine-tuning stage. For summarization and medical dialog models, we identified memorization ratios of 20.7% and 19.6%, respectively. These high rates could lead to privacy violations or copyright issues in applications like academic sum-

---

[1]https://www.elastic.co/elasticsearch/

[2]https://pan.webis.de/clef14/pan14-web/text-alignment.html

[3]To validate paraphrasing, a RoBERTa model and NER compute confidence scores $p$. Cases with $p < 0.5$ are low-confidence paraphrasing, while $p > 0.5$ are high-confidence. Both types are reported in the results.

[4]We attached the description of all models and dataset used in Appendix C

Table 1: Memorization Effects of Open-sourced LLMs Fine-tuned on Various Tasks.

| Task | Dataset | Source Model | Memorization Ratio | Verbatim | Idea | Paraphrase ($p > 0.5$) | Paraphrase ($p < 0.5$) |
|---|---|---|---|---|---|---|---|
| Summarization | CNN/Daily Mail | Bart_Large | 20.7% | 1.3% | 0% | 9.8% | 9.6% |
| Medical Dialog | ChatDoctor | BioGPT | 19.6% | 0.1% | 3.5% | 7.8% | 8.2% |
| Extractive QA | SQuAD_v2 | T5_large | 0.1% | 0% | 0% | 0% | 0.1% |
| Abstractive QA | Race | T5_large | 0.3% | 0% | 0% | 0.2% | 0.1% |
| Translation | WMT_19 | FSMT | 0% | 0% | 0% | 0% | 0% |
| Sentiment Classification | IMDB | T5-base | 0% | 0% | 0% | 0% | 0% |

marization and healthcare chatbots. Furthermore, the amount of memorization varies across tasks, displaying clear task-specific patterns. Models fine-tuned for summarization and medical dialog exhibit high memorization, while models for remaining tasks show much lower memorization. These observations motivate further in-depth analysis to validate the observed task-specific memorization behavior and explain potential underlying factors in Section 4. Section 5 introduces attention score density as a potential indicator to quantify and compare memorization tendencies across diverse tasks. Finally, Section 6 investigates multi-task fine-tuning.

## 4 FINE-TUNING MEMORIZATION EFFECTS ARE TASK-SPECIFIC

Our preliminary study initially demonstrates that fine-tuned models' memorization may be related to their fine-tuning tasks. However, other factors such as the fine-tuning datasets or model architectures could also impact the memorization effect. Therefore, in Section 4.1, we systematically control the impact of such variables to precisely explore the relation between fine-tuning task and the memorization effect. In Section 4.2, via constructing conceptual models, we discuss the underlying difference between these fine-tuning tasks which potentially influence the memorization of LMs. In our fine-tuning process, we make sure our fine-tuned models have satisfied performance on downstream tasks and report the performance in Appendix D. Besides, we add ablation studies for different decoding methods in Appendix B.2 , prefix lenghs in Appendix B.3 and report the memorization ratio of T5-base model in Appendix B.1

Table 2: Memorization Effects of T5-base Fine-tuned on Various Tasks.

| Task | Dataset | Source Model | Memorization Ratio | Verbatim | Idea | Paraphrase ($p > 0.5$) | Paraphrase ($p < 0.5$) |
|---|---|---|---|---|---|---|---|
| Summarization | Multi_news | T5_base | 22.33% | 4.23% | 0.65% | 6.23% | 11.22% |
| Medical Dialog | HealthCareMagic | T5_base | 8.27% | 0.02% | 1.41% | 1.75% | 5.09% |
| Extractive QA | SQuAD_v2 | T5_base | 0.15% | 0.04% | 0.00% | 0.05% | 0.06% |
| Translation | WMT_16 | T5_base | 0.00% | 0.00% | 0.00% | 0.00% | 0.00% |
| Sentiment Classification | IMDB | T5_base | 8.00% | 0.04% | 0.30% | 0.17% | 0.29% |

### 4.1 FINE-TUNING WITH FIXED BASE LM AND DATASET

**Fine-tuning on T5-base LM.** To eliminate the potential impact from the base LM, we conduct experiments to fine-tune the same pre-trained T5-base model for different fine-tuning tasks. Note that we treat all the tasks as generative tasks (like instruction-tuning) and do not add any additional architecture like MLP. Details of the datasets and memorization results are presented in Table 2. The results clearly show substantial memorization effects for summarization tasks (22.3%) and medical dialogue (8.27%), while much fewer cases occurred for reading comprehension (0.15%), translation (0.0%), and sentiment classification (0.8%). These observations suggest that with the same base LM architecture, fine-tuning memorization still demonstrates a strong task disparity.

**Fine-tuning on RentTheRunway with T5.** In this experiment, we further eliminate the potential impact of the difference among fine-tuning datasets. To achieve this goal, we investigate the memorization of different tasks fine-tuned on the same base LM with the same dataset. Specifically, we chose the RentTheRunway dataset Misra et al. (2018) for fine-tuning and used the T5-base model as the pre-trained starting point. The RentTheRunway dataset contains self-reported clothing fit feedback from customers along with additional metadata. Each product has multiple attributes, including customer reviews, ratings, review summaries, review dates, etc. In this experiment, we used the customer reviews as input and the review summaries as output to fine-tune a summarization

model. At the same time, we divided the ratings into positive and negative labels to use as outputs for training a sentiment classification model. In this setting, the two models have different task objectives but identical inputs and pre-trained starting points. The memorization performance of these two models is shown in Figure 1. The results demonstrate that the summarization model exhibited higher memorization rates compared to the classification model. This finding validates that the task objective impacts the memorization of fine-tuned models.

## 4.2 UNDERSTANDING THE MEMORIZATION DISPARITY AMONG TASKS

In this subsection, we aim to further investigate the question: *why do different fine-tuning tasks present different memorization behaviors?* Intuitively, for language tasks such as sentiment analysis or extractive QA, only a few words or sentences are enough for the model to complete the task (e.g. words expressing sentiment, the answer to the question). Thus the model only needs to learn specific key features and is less likely to memorize the data species. However, the tasks such as summarization and dialog, require the model to understand the whole passage, which could boost memorization. In other words, the memorization behavior might be closely related to the information needed to fulfill certain language tasks. To give a more detailed analysis, in the following, we will provide a conceptual discussion based on a sparse decoding model, which is a popular model for modeling text and vision data. (Arora et al., 2015; 2018; Olshausen & Field, 1997; 2004)

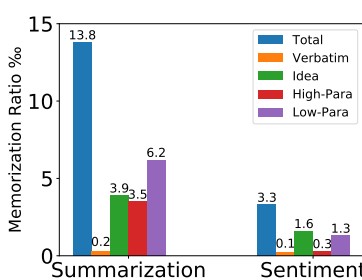

Figure 1: Memorization of T5 in RentTheRunway.

**Sparse coding model.** Suppose that an observed text data is denoted as $Z$, and $Z \in \mathbb{R}^{d \times D}$ where $D$ is the sequence length and $d$ is the length of the embedding. The basic assumption of sparse coding is that the data $Z$ comes from combinations of a few hidden features. Hence, we use $X$ to represent these essential features and we have the relation:

$$Z = UXV \tag{1}$$

where $X \in R^{k \times K}$, $U \in \mathbb{R}^{d \times k}$, $V \in \mathbb{R}^{K \times D}$ , $k \leq d$, $K \leq D$. Each column of $U$ is a unit vector and orthogonal with each other. Each row of $V$ is a unit vector and orthogonal with each other. Therefore, each element in $Z$ is a linear combination of the elements from $X$. Our assumption is a simplified version of the sparse coding model without noise. We also modify the original 1D feature $X$ in sparse coding into 2D for our model.

**Task Complexity.** Under sparse coding, we further assume the model's targeted output can be fully expressed by a linear transformation of $X$. However, different fine-tuning tasks may differ in *how much information of the input is needed by the task*. Below, we give two perspectives to illustrate why "complex tasks" may have more memorization.

First, the number of parameters to connect $Z$ with the final target is related to the information needed by the task. Consider we have a *"simple task"* (like sentiment analysis) where the model output is only one scalar (preference). For example, this scalar is decided by a linear combination of $X$, which is $a^\top X b$, where $a \in \mathbb{R}^k$ and $b \in \mathbb{R}^K$. Thus, the loss function for a sentiment classification model $f_{cls}$ can be defined as:

$$l(f_{cls}(Z), a^\top X b). \tag{2}$$

If we further assume the loss functions as square loss, the best solution of $f_{cls}$ is: $f_{cls}(Z) = (a')^\top Z b'$ where $a^\top U^\top := a'$ and $V^\top b := b'$. It means that the model only needs to learn two vector parameters $a'$ and $b'$. Nevertheless, for some tasks such as summarization, the output text is desired to contain all key information from the input passage $Z$. Thus, the target output can be expressed by $X$. We consider the following loss for a summarization model $f_{sum}$:

$$l(f_{sum}(Z), X). \tag{3}$$

The output $f_{sum}(Z)$ therefore contains all information about $X$, which we denote as *"complex task"*. Considering square loss, the best solution of $f_{sum}$ is $f_{sum}(Z) = U^\top Z V^\top$. Comparing the above two tasks, for simple tasks like classification, the model just requires a small amount of information and needs to learn minimal $(d + D)$ values in $a', b'$, while for complex tasks like

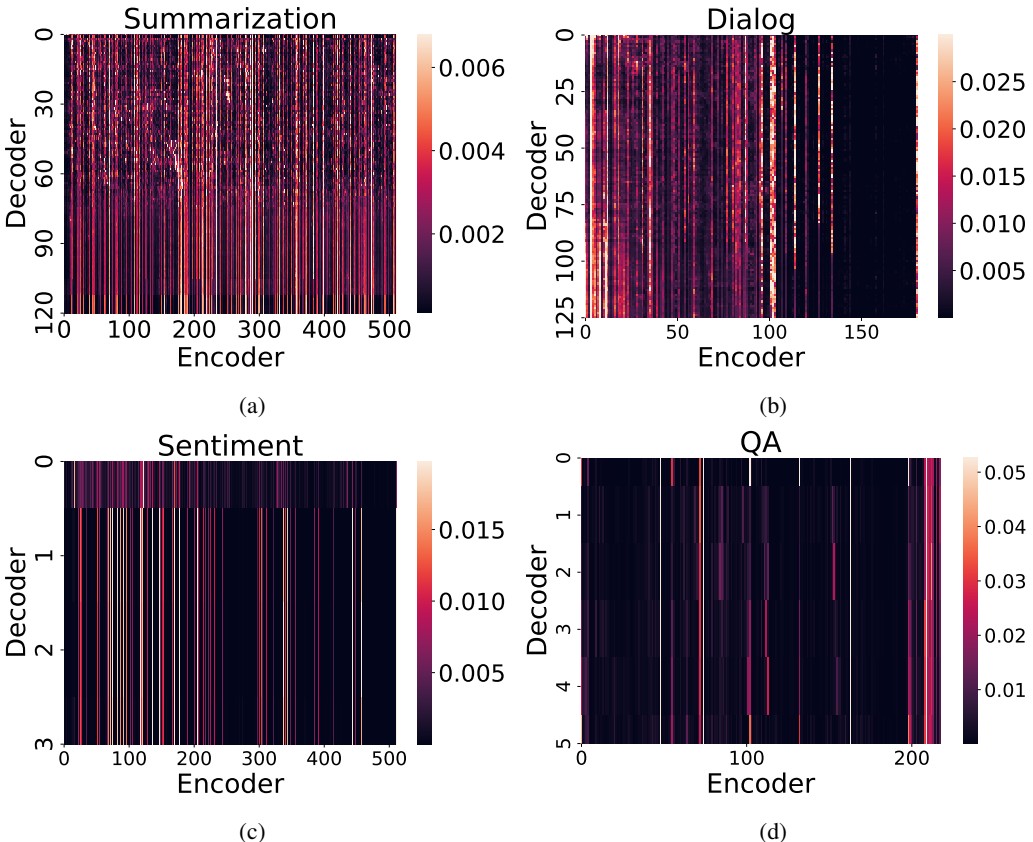

Figure 2: Decode-encoder attention heatmaps on (a) summarization, (b) dialog, (c) sentiment, and (d) QA

summarization, the model needs to learn $(dk + DK)$ parameters of $U' = U^\top, V' = V^\top$. Intuitively, the model that learns more information from training data tends to have larger memorization. An alternative explanation for this is that the expression in Equation 3 makes model inversion attack possible. As the model learns $U', V'$, the attacker can conduct a model inversion attack via $Z = U'^T X V'^T$, which means one can use $f(Z)$ to recover the input data $Z$. This means the information of data $Z$ is fully encoded in the neural network.

Second, the sparsity of the learned matrix $(U, V)$ or vectors $(a', b')$ may also vary, indicating different amounts of information needed and leading to different complexity of the task. For example, in sentiment classification, what the network actually learns depends on the sparsity of $b'$. If $b'$ is sparse, it means that we can simply pick several tokens from the sequence and determine the class.

## 5 ATTENTION SCORES AS INDICATORS

In the last section, we empirically validate the memorization disparity among tasks. Meanwhile, via a sparse coding model, we also conceptually show that this disparity may relate to the information needed for the task. However, these analyses are still impractical to quantify tasks as high or low memorization. In this section, we unveil that attention density is highly related to memorization. Consequently, we introduced CAD score to measure the density of attention distribution and we found that it positively correlated with the memorization ratio of the task. Thus we claim that it can be used as a potential memorization indicator. We also conduct theoretical and empirical analyses to understand this correlation.

### 5.1 ATTENTION DISTRIBUTIONS

As discussed above, the memorization disparity is possibly related to the essential information needed by the task. Meanwhile, the attention distribution in the Transformer captures the contri-

bution of each token's information to completing the task. This prompted us to consider whether attention score could be a good indicator. Thus, for the fine-tuned models in Table 2, we generate their attention score heatmaps in Figure 2. It shows the attention distribution of the last decoder-encoder attention block in each model. We also visualize different layers of decoding-encoding attention scores in Appendix F.2 and the attention scores of T5-base model on Appendix F.1 The horizontal axis represents input tokens, while the vertical axis indicates output tokens. We use different colors to show the decoder-encoder attention (averaged attention of multi-heads) between each pair of input and output tokens. Each horizontal line is an attention distribution of one output token on its input tokens. The sum of each attention distribution is 1. Every heatmap uses the average attention of 10 samples, and each sample is padded to the longest length in the batch and truncated to at most 512 tokens. The heatmaps in Figure 2 reveal clear differences in attention score patterns across tasks. For high-memorization tasks like summarization and dialog, attention scores are evenly distributed across input tokens. In contrast, for low-memorization tasks like sentiment classification and extractive QA, attention is concentrated on a few positions while minimal for other positions. The observed patterns suggest the information needed to successfully complete each task varies. Even attention score distribution for summarization and dialog implies that models must attend to every input detail, increasing the likelihood of input memorization. Concentrated scores for sentiment classification and extractive QA indicate that only key information is required, reducing the tendency to memorize training data specifics.

## 5.2 CORRECTIVE ATTENTION DISPERSION

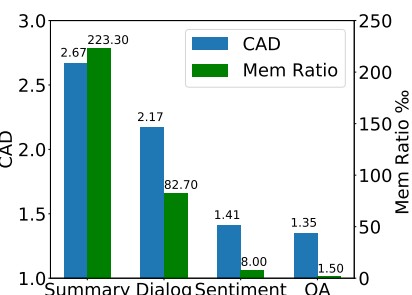

Figure 3: CAD vs Memo Ratio

Based on this observation, we propose Corrective Attention Dispersion (CAD) to quantify the dispersion degree of the attention score by analogizing the entropy of discrete probabilistic distributions. For probabilistic distribution, entropy can measure the uncertainty of the random variable and describe its dispersion:

$$E_{\text{entropy}}(\mathcal{X}) = \sum_{i=1}^{N} -p_i \log\left(p_i\right),$$

where $N$ is the number of states of random variable $\mathcal{X}$. If the distribution is dispersed, the entropy will be low, which means the states of $\mathcal{X}$ are more uncertain. For the dispersion of attention score, we use CAD as an analogy of entropy. Each horizontal line in the heatmap is a distribution of attention scores on input tokens. Each input token is seen as a state, while the attention score between an input token and the output token of that line is its probability. In this way, CAD can measure the dispersion degree of the attention score. The length of each heatmap to obtain a one-dimensional discrete distribution is equal to the input length $l_t$. However, the number of states will influence the entropy. For example, the entropy of $n$-state discrete uniform distribution is $\log n$. This is because more states mean more uncertainty. The number of input tokens is usually different. Thus, we propose CAD to align the entropy between the distributions of samples with different numbers of input tokens:

$$E_{\text{CAD}}(\mathcal{A}) = \sum_{i=1}^{l} -\left[\frac{a_i}{r} \log\left(\frac{a_i}{r}\right)\right] r = \sum_{i=1}^{l} -a_i \log\left(\frac{a_i}{r}\right),$$

where $r = L/l_t$, $L$ is the corrected number of input tokens, $l_t$ is the original number of input tokens, $a_i$ is the attention score at position i. CAD can be seen as a correction to split or pad the attention score into $r$ new tokens and align the length of input tokens to the same $L$ for all the samples from different tasks.

In this work, we use $L = 10$ and report the results in Figure 3. From the figure, we note that summary > dialog > sentiment > QA in terms of CAD, which aligns with the observed memorization behavior. The high CAD of the attention of summarization and dialog indicates they have more dense attention distributions and their dependence on all input details. Whereas the lower CAD of sentiment and QA suggests their concentration on only key semantic features. The results show the CAD of attention distribution can align well with memorization.

## 5.3 UNDERSTANDING THE RELATION BETWEEN ATTENTION SCORES AND MEMORIZATION

To understand why attention scores can be used as an indicator for memorization, we provide some theoretical intuition on the relation between attention scores and information needed for the task. We first use the classification task to explain the relation between attention scores and information density. Then we extend the intuition to discuss complex tasks.

**Attention score and memorization in classification**    We still consider the sparse coding model mentioned in Eq.1, and continue to use the notations of Section 4.2. We use the classification task for simplicity. As mentioned, for classification tasks, the best solution of Eq.2 is $f_{cls}(Z) = (a')^\top Z b'$, where $a^\top U^\top := a'$ and $V^\top b := b'$. In classification, whether a task is more complex or not depends on the sparsity of $b'$, and we justify that the **sparsity of $b'$ directly affects the attention score pattern.**    We then mathematically define the neural network architecture. To ease the derivation, we consider

$$f(Z) = W^V Z \cdot \text{softmax}\left((W^K Z)^\top (W^Q Z)\right), \tag{4}$$

with $W^V, W^K, W^Q$ all in $\mathbb{R}^{d \times d}$. The softmax operation is conducted column-wise. Since the output $f(Z)$ is a matrix rather than a scalar, for the classification task, we further multiply two vectors on the two sides of $f(Z)$ to get output scalar $y'$, i.e., $v_1^\top f(Z) v_2 \in \mathbb{R}$, and $v_1$ and $v_2$ can be either trainable or arbitrary. As mentioned in Section 4.2, the target of the classification task can be represented as $y = a^\top X b$. The loss term of Eq.2 can then be written as:

$$l(v_1^\top f(Z) v_2, a^\top X b). \tag{5}$$

Aligning the neural network output $v_1^\top f(Z) v_2$ with $a^\top X b$, it is easy to see that to better reduce the loss value, we need softmax $\left((W^K Z)^\top (W^Q Z)\right) v_2 \in \mathbb{R}^D$ better aligned with $b'$. As a result, when measuring the effect of the input $Z$ on $v_1^\top f(Z) v_2 \in \mathbb{R}$, e.g., Figure 2c, the weighted pattern softmax $\left((W^K Z)^\top (W^Q Z)\right) v_2$ has a similar sparsity as $b'$. Recall that $b'$ is the task-specific vector the model needs to learn, thus the analysis above suggests that **the attention score[5] has a similar sparsity pattern as the sparsity of the information the model needs to learn.**

**Complex tasks**    For more complex tasks, the simplistic single-layer single-head attention analysis as the above is not enough to handle it, and we need to use a larger architecture. Intuitively, with more features to learn in the task, the architecture will be more likely to memorize each feature comprehensively. We identify two key drivers of this behavior. First, each output token relies on information distributed across multiple input tokens. As shown in Figure 2a, each row has multiple high attention scores across different input tokens. Second, each output token often exhibits selectivity for a different subset of input tokens, leading to divergence in attention distributions across rows in Figure 2a. To conclude, these two factors may result in dense heatmap patterns compared to the concentrated heatmaps of simpler tasks.

## 6 MULTI-TASK FINE-TUNING CAN MITIGATE MEMORIZATION

Beyond fine-tuning on a single task, multi-task fine-tuning has been also widely utilized, like (Longpre et al., 2023) (Wei et al., 2021)(Ouyang et al., 2022) Chung et al. (2022). Multi-task fine-tuning typically leverages a pre-trained language model and continues training across several downstream tasks at a time. Multi-task fine-tuning is a promising approach to enhance the capabilities and generalizability of pre-trained language models for diverse downstream applications. In this section, we investigate the effects of multi-task fine-tuning on model memorization.

We first examine the memorization behavior of FLAN-T5 Chung et al. (2022), an enhanced version of T5 that has been fine-tuned on over 1,000 additional tasks beyond the original T5 model. We select several popular tasks and datasets that were utilized to fine-tune FLAN-T5 and examine the memorization behavior of FLAN-T5. The results are presented in Table 3. On the one hand, similar to single-task fine-tuning, multi-task fine-tuning also presents memorization effects and demonstrates similar task disparity. For instance, we can see the memorization of summarization is obviously higher than other tasks like classification and reading comprehension. On the other hand, the memorization ratio for high-memorization tasks

---

[5] Note that the attention matrix softmax $\left((W^K Z)^\top (W^Q Z)\right)$ itself is $\mathbb{R}^{D \times D}$ and is not aggregated for the output value, and softmax $\left((W^K Z)^\top (W^Q Z)\right) v_2$ is the final aggregated attention.

Table 3: Memorization of FLAN_T5

| Task | Sub_task | Dataset | Total | Verbatim | Idea | Paraphrase ($P < 0.5$) | Paraphrase ($P > 0.5$) |
|------|----------|---------|-------|----------|------|------------------------|------------------------|
| Summarization | Summarization | Multi_news | 2.96% | 0.73% | 0.42% | 0.51% | 1.81% |
| | | CNN_Daily | 0.92% | 0.01% | 0.16% | 0.10% | 0.75% |
| | | Xsum | 0.46% | 0.01% | 0.04% | 0.08% | 0.41% |
| Reading Comprehension | Extractive | SQuAD_v2 | 0.06% | 0.03% | 0.00% | 0.00% | 0.03% |
| | | adversarial_qa | 0.00% | 0.00% | 0.00% | 0.00% | 0.00% |
| | Abstractive | duorc_SelfRC | 0.02% | 0.00% | 0.00% | 0.00% | 0.02% |
| | | duorc_ParaRC | 0.09% | 0.00% | 0.05% | 0.00% | 0.04% |
| | Multiple Choice | boolq | 0.13% | 0.00% | 0.02% | 0.04% | 0.11% |
| Classification | Classification | AG_news | 0.03% | 0.01% | 0.00% | 0.01% | 0.02% |
| | | dbpedia_14 | 0.03% | 0.00% | 0.01% | 0.01% | 0.02% |
| Translation | Translation | WMT_14 | 0.00% | 0.00% | 0.00% | 0.00% | 0.00% |
| | | WMT_16 | 0.00% | 0.00% | 0.00% | 0.00% | 0.00% |
| NLI | NLI | super_glue/cb | 0.00% | 0.00% | 0.00% | 0.00% | 0.00% |
| | | super_glue/wsc | 0.00% | 0.00% | 0.00% | 0.00% | 0.00% |

is substantially lower than that of single-task fine-tuned models. For example, the memorization ratio of summarization (2.96% on multi_news and 0.92% on CNN_Daily ) in multi-task fine-tuning is much lower compared to that of single-task fine-tuning shown in Table 2 (22.33% on Multi_news) and Table 1 (20.7% on CNN_Daily). This observation suggests that multi-task fine-tuning has the potential to mitigate the memorization of high-memorization tasks.

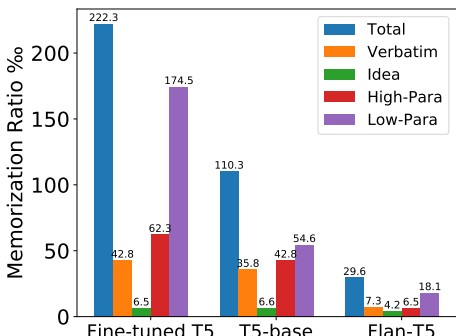

Figure 4: Memorization Comparison of Fine-tuned T5, T5-base, and FLAN-T5 in Summarization on Multi_News

To further investigate the impact of multi-task fine-tuning on memorization of high-memorization tasks, we compare the memorization behavior of T5-base (Pre-trained), fine-tuned T5 (single-task fine-tuning), and FLAN-T5 (multi-task fine-tuning) as shown in Figure 4. Note that we choose summarization as the high-memorization task and test on the Multi_News dataset in this study. The high memorization ratio of T5-base (11.03%) suggests the overlap between the fine-tuning dataset and the pre-training dataset. Notably, the memorization ratio is increased markedly after single-task fine-tuning (22.23%) compared to T5-base (11.03%), indicating the single-task fine-tuning process causes the model to memorize more of the training data. On the other hand, the memorization ratio of the multi-task fine-tuned FLAN-T5 (2.96%) is substantially lower than that of T5-base (11.03%) and fine-tuned T5 (22.23%). In particular, the memorization ratio of FLAN-T5 is approximately 5 times lower than T5-base and 10 times lower than fine-tuned T5. These observations further validate that multi-task fine-tuning has the potential to mitigate memorization of high-memorization tasks. A plausible explanation for this observation is the enhanced generalization capability of the model in a multi-task setting. Exposure to a broader array of tasks and data appears to facilitate this improved generalization. Consequently, the model's ability to execute tasks is not solely dependent on memorization but also on its refined generalization skills.

## 7 CONCLUSIONS

In this paper, we first extensively investigated the memorization behavior of fine-tuned language models across various tasks. Utilizing an automatic detection pipeline, we evaluated numerous tasks and datasets. We provide an understanding of this memorization disparity among tasks based on a sparse coding theory. Further analysis revealed a strong correlation between attention scores and memorization. Finally, we examined memorization effects on multi-task fine-tuned language models and found that multi-task fine-tuning paves a potential strategy to mitigate memorization.

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

# A    IMPLEMENTATION DETAILS OF EVALUATION PIPELINE

Here we introduce the evaluation tools and pipelines in detail. The evaluation contains 2 processes, search and detection. Such pipeline was first proposed and adopted by Lee et al. (2023)

**Search**    In the search stage, a local search engine called Elasticsearch is implemented. Elasticsearch is a distributed, RESTful search and analytics engine built on top of the open-source Lucene library. Elasticsearch utilizes the Okapi-BM25 algorithm, a popular bag-of-words ranking function. It enables users to store, search, and analyze large volumes of data quickly and in near real-time. In our experiment, we choose $n = 10000$ samples of each dataset to be evaluated. In the evaluation process, we first divide the input of evaluated data $x_i$ into prefix $p_i$ and suffix $s_i$. We then load all suffix $\{s_i\}_{i=1}^n$ into Elasticsearch and input $\{p_i\}_{i=1}^n$ into the model to get $f(p_i)$ and use $\{p_i\}_{i=1}^n$ as the query documents. We used $K = 10$ in our experiments. We typically choose the prefix-length $k$ as 50 tokens. (For some datasets whose input sentences are shorter than 50 tokens, we use 15 tokens instead. )

**Detection**    We utilize PAN2014 plagiarism detection to identify memorized cases and obfuscation types. Their method includes (1) preprocessing text; (2) identifying obfuscation types; (3) seeding to find candidate pairs via sentence similarity; (4) extension by clustering similar fragments; and (5) filtering out overlaps. They transform sentences into TF-IDF vectors and calculate similarity using dice and cosine measures, with adaptive parameters selected by testing on the obfuscation corpus. Here we set the minimal match threshold as at least 50 characters(approximately 20 tokens). We also utilize additional validation steps after retrieving paraphrased text segments as Sanchez-Perez et al. (2015). The post-processing involves chunking segments into sentences using NLTK's tokenizer, then applying a RoBERTa-based paraphrase identification model and Named Entity Recognition (NER) on the sentences. Specifically, we check sentence pairs - if any pair has a paraphrase detection probability score between 0.5 and 0.99, we accept it as high-confidence paraphrasing, otherwise, we identify it as low-confidence paraphrasing.

**Memorization Type**    Here we will distinguish 3 types of memorization. First, verbatim memorization means exact copies of words or phrases without transformation. In the cases of paraphrase and idea memorization, the output is not identical to the original text but shares similar meanings. While paraphrase plagiarism focuses on sentence-to-sentence transformation, idea plagiarism involves summarizing the key points of a larger text segment into a more condensed form (or expanding it)

In practice, the PAN2014-detection starts by identifying closely matched short document fragments (referred to as 'seeds') and then expands these seeds into longer text segments. This is achieved by clustering these fragments based on their separation and employing a 'maxgap' threshold parameter to form coherent clusters. They experimentally find out the most suitable threshold for different plagiarism datasets so that those parameters could be used for the detection of a specific type of memorization. In other words, each memorization case will be counted only once and there will not be overlapping across different categories.

# B    ABLATION STUDIES

## B.1    COMPARISON WITH T5-BASE

Table 4: Memorization Effects of T5-base models

| Task | Dataset | Memorization Ratio | Verbatim | Idea | Paraphrase ($p > 0.5$) | Paraphrase ($p < 0.5$) |
|---|---|---|---|---|---|---|
| Summarization | Multi_news | 11.03% | 3.58% | 0.66% | 4.28% | 5.46% |
| Dialog | HealthCareMagic | 1.60% | 0.03% | 1.04% | 0.11% | 0.42% |
| Sentiment | IMDB | 0.78% | 0.05% | 0.37% | 0.16% | 0.20% |
| QA | Squad | 0.09% | 0.02% | 0.00% | 0.01% | 0.05% |
| Translation | WMT_16 | 0.00% | 0.00% | 0.00% | 0.00% | 0.00% |

Table 5: Differential Memorization Ratios of T5-base Across Tasks.

| Task | Dataset | Memorization Ratio Difference |
|------|---------|------------------------------|
| Summarization | Multi_news | 11.30% |
| Dialog | HealthCareMagic | 6.67% |
| Sentiment | IMDB | 0.20% |
| QA | Squad | 0.06% |
| Translation | WMT_16 | 0.00% |

To eliminate the impact that the fine-tuning data may also present in the pre-training data, we test the memorization ratio of T5-base without fine-tuning in Table 4, and calculate the memorization difference after fine-tuning in Table 5. From the results we can find that a part of the fine-tuning data may also be present in the pre-training data, as T5-base also shows some cases of memorization of the fine-tuning data. However, after fine-tuning, we can see that the fine-tuned model's memorization for summarization and dialogues improves significantly. The model has memorized a large number of fine-tuning data samples during the fine-tuning stage. However, for other tasks such as sentiment classification, the memorization ratio does not change much, meaning that the model does not remember much information from the fine-tuned data.

## B.2 DECODING METHODS

Table 6: Memorization of Finetuned_T5 with Various Decoding Methods.

| Task | Dataset | Decoding | Memorization Ratio | Verbatim | Idea | Paraphrase ($P < 0.5$) | Paraphrase ($P > 0.5$) |
|------|---------|----------|--------------------|----------|------|------------------------|------------------------|
| Dialog | HealthCareMagic | Top-K | 5.76% | 0.05% | 0.38% | 0.90% | 4.43% |
| | | Top-p | 7.26% | 0.06% | 0.48% | 1.35% | 5.37% |
| | | Temp | 3.72% | 0.02% | 0.18% | 0.58% | 2.94% |
| | | Greedy | 8.27% | 0.02% | 1.41% | 1.75% | 5.09% |
| Sentiment | IMDB | Top-K | 1.02% | 0.01% | 0.13% | 0.18% | 0.70% |
| | | Top-p | 1.08% | 0.01% | 0.12% | 0.22% | 0.73% |
| | | Temp | 0.89% | 0.01% | 0.07% | 0.19% | 0.62% |
| | | Greedy | 0.80% | 0.04% | 0.30% | 0.17% | 0.29% |
| Summarization | Multi_news | Top-K | 10.80% | 2.54% | 0.34% | 1.94% | 5.98% |
| | | Top-p | 13.57% | 4.07% | 0.54% | 2.26% | 6.70% |
| | | Temp | 5.82% | 1.28% | 0.23% | 0.83% | 3.48% |
| | | Greedy | 22.33% | 4.23% | 0.65% | 6.23% | 11.22% |

We conduct ablation studies on different decoding methods in Table 6. From the results, we can find that:

- For high-memory tasks such as summarization and dialogue, sampling can reduce the memorization ratio and change the category distribution of memory samples.

- For low-memory tasks such as emotion classification, sampling does not reduce the memorization ratio (even increases it in some way), and the changes are not very profound.

- Irrespective of the decoding methodology employed, **a pronounced disparity in memorization across different tasks persists.** This suggests an inherent task-specific propensity towards memorization that is not substantially mitigated by variations in sampling techniques.

## B.3 PREFIX LENGTHS

Here we change different prefix lengths of inputs and report the results in table 7. We include 2 high-memorization tasks(summarization and dialog) and 1 low-memorization task (sentiment classification). From the results we can observe that:

Table 7: Memorization of Finetuned_T5 with Varying Prefix Lengths.

| Task | Dataset | Prefix length | Memorization Ratio | Verbatim | Idea | Paraphrase $(p > 0.5)$ | Paraphrase $(p < 0.5)$ |
|---|---|---|---|---|---|---|---|
| Summarization | Multi_news | 10 | 12.25% | 1.74% | 2.85% | 0.88% | 6.78% |
| | | 30 | 20.68% | 7.07% | 1.41% | 3.05% | 9.15% |
| | | 50 | 22.33% | 4.23% | 0.65% | 6.23% | 11.22% |
| | | 100 | 29.66% | 10.61% | 0.79% | 4.27% | 13.99% |
| Dialog | HealthCareMagic | 10 | 6.28% | 0.03% | 1.94% | 0.85% | 3.46% |
| | | 30 | 7.76% | 0.04% | 1.28% | 1.72% | 4.72% |
| | | 50 | 8.27% | 0.02% | 1.41% | 1.75% | 5.09% |
| Sentiment | IMDB | 10 | 1.37% | 0.00% | 1.12% | 0.06% | 0.19% |
| | | 30 | 1.18% | 0.01% | 0.51% | 0.15% | 0.51% |
| | | 50 | 0.80% | 0.04% | 0.30% | 0.17% | 0.29% |
| | | 100 | 1.39% | 0.05% | 0.23% | 0.33% | 0.78% |

- **The length of prefix tokens can affect memorization.** The length of prefix tokens does indeed impact memorization. Specifically, for summarization and Dialog tasks, the memorization ratio generally increases with the length of the prefix. This finding aligns with previous research on pre-trained memorization. However, for sentiment classification, changing the prefix does not result in significant changes, and increasing the prefix length does not necessarily lead to an increase in the memorization ratio.

- **The task disparity still exists when using different prefixes.** Furthermore, it is worth noting that despite the influence of different prefixes on memorization, there still exists a noticeable disparity in memorization across tasks. Therefore, our conclusion remains even using different prefixes.

## C DATASET AND MODEL USED

**Datasets** Here we introduce the datasets we used in preliminary experiments and model fine-tuning. For the summarization task. We consider CNN/Daily Mail in the preliminary study. This is a widely used dataset for text summarization. It contains 287k rows of data for training, and we use 10k data to evaluate memorization. For our fine-tuning, we use **Multi-News** dataset for fine-tuning. This dataset contains news articles and professional human-written summaries of these articles from the site newser.com. We use a 45k train set for finetuneing and 5.62k test set for model performance evaluation. We use the **IMDB** dataset for the binary sentiment classification task. IMDB dataset contains movie reviews and corresponding sentiment labels divided into 25k training data and 25k test data. For the dialogue task, we conduct the preliminary study on ChatDoctor dataset. Our dialogue model is fine-tuned with a 112k row dataset from **HealthcareMagic** online doctor consultation data. We select 100k for the training set and 12k for the test set. For the translation task, the preliminary stud use **WMT19** while we select an English-to-German subset of **WMT16**, with 450.87k training set and 3k test set. For extractive QA task, we use Stanford Question Answering Dataset (**SQuAD** v2), which consists of questions on a set of Wikipedia articles, where the answer to every question is a segment of text. The training set contains 130k rows and the test set contains 11.9k rows. **RentTheRunway** is a subset of Recommender Systems and Personalization Datasets. This dataset contains cloth fitting review data, including reviews, summary of reviews, and ratings. We use 111k as training data and 12k as test data. The review and summary data is used to fine-tune the summarization model and the rating data is used to fine-tune the binary sentiment classification task where ratings larger than 5 are viewed as positive.

**Models** In the preliminary study, we consider Bart-Large from Bart family, T5-base and T5-large from T5 family, FSMT (FairSeq MachineTranslation), and BioGPT. For our self-fine-tuned models, we select T5-base architecture from the T5 family for all experiments.

## D    PERFORMANCE OF SELF-FINE-TUNED MODELS

We finetune the T5-base model to achieve better or comparable performance with the Google fine-tuned public model FLAN-T5. In Table8, we show the performance of the summarization task. Our fine-tuned model achieves a similar rouge score with FLAN-T5. In Table 9, We show that the accuracy of our model is better than FLAN-T5 regarding binary sentiment classification. For Dialogue task, Our model performance much better than FLAN-T5 as shown in Table10. Similarly, for translation task, our fine-tuned model has a much higher BLEU score than FLAN-T5, as shown in Table11. For Extractive question answering task, we fine-tune the model in a sequence-to-sequence learning form while we evaluate the exact match of the answer term. Results are shown in Table 12. For the RentTheRunway fine-tuning experiment, we present the results in Table 13 and Table 14.

Table 8: Summarization

| Dataset | Model | Rouge1 | Rouge2 | RougeL | RougeLSum |
|---|---|---|---|---|---|
| multi_news | FLAN-T5-Base | 0.291 | 0.098 | 0.237 | 0.237 |
| multi_news | Our fine-tuned T5-Base | 0.298 | 0.103 | 0.201 | 0.201 |

Table 9: Sentiment Classification

| Dataset | Model | Accuracy(%) |
|---|---|---|
| IMDB | FLAN-T5-Base | 93.56 |
| IMDB | Our fine-tuned T5-Base | 94.64 |

Table 10: Dialogue

| Dataset | Model | Rouge1 | Rouge2 | RougeL | RougeLSum |
|---|---|---|---|---|---|
| HealthCareMagic | FLAN-T5-Base | 0.0546 | 0.0058 | 0.0389 | 0.0389 |
| HealthCareMagic | Our fine-tuned T5-Base | 0.298 | 0.103 | 0.201 | 0.201 |

Table 11: Translation

| Dataset | Model | BLEU | brevity penalty | length ratio | translation length |
|---|---|---|---|---|---|
| WMT16 de-en | FLAN-T5-Base | 5.62 | 0.943 | 1.14 | 22.71 |
| WMT16 de-en | Our fine-tuned T5-Base | 20.55 | 0.947 | 1.04 | 21.30 |

Table 12: Question Answering

| Dataset | Model | Exact Match(%) |
|---|---|---|
| SQuAD v2 | FLAN-T5-Base | 0 |
| SQuAD v2 | Our finetuned T5-Base | 44 |

Table 13: Multi Task Trained with RentTheRunway

| Dataset | Model | Rouge1 | Rouge | RougeL | RougeSum |
|---|---|---|---|---|---|
| Summary | FLAN-T5-Base | 0.1743 | 0.0436 | 0.1598 | 0.1598 |
| Summary | Our finetuned T5-Base | 0.1743 | 0.0436 | 0.1598 | 0.1598 |

Table 14: Multi Task Trained with RentTheRunway

| Dataset | Model | Accuracy(%) |
|---|---|---|
| Sentiment Classification | Flan T5-base | 86.60 |
| Sentiment Classification | Our fine-tuned T5-base | 98.07 |

# E    MEMORIZED EXAMPLES

We present memorization examples of verbatim, paraphrase and idea plagiarism of different models.

Table 15: Examples of Memorization Cases. Duplicated texts are highlighted with yellow marks. Personally identifiable information (PII) and other words that may lead to privacy concern in generated text are masked as red.

| Type | Machine-Written Text | Training Text |
|------|----------------------|---------------|
| Paraphrase | At least 10 people and two attackers were killed in Tuesday's attack against the luxurious Corinthia Hotel in Tripoli, Libya, a spokesman for a security division of the Ministry of Interior in Tripoli said. Five foreigners – one American (***Summarization: CNN Daily, Bart Large***) | On January 27, gunmen claiming to be affiliated with ISIS attacked the Corinthia Hotel in Tripoli, Libya, which is favored by government officials and foreigners. They killed 10 people after storming into the lobby and firing guns at hotel guests. Five of the victims were foreigners, one an American. |
| Paraphrase | – Argentine President Cristina Fernandez de Kirchner was told to take a month off work after doctors diagnosed her with a subdural hematoma. (***Summarization: CNN Daily, Bart Large***) | ...hematoma and said she needed to take a month off of work. |
| Paraphrase | I am a 20-year-old guy 20 years old. I have been (***harassment word***) for a long time. (***Dialogue: HealthCareMagic, Finetuned T5***) | ...20 years old i have been. (***harassment word***) regularly for past 5 years |
| Paraphrase | The Chargers responded with a 1-yard TD run by RB LaDainian Tomlinson. (***Abstractive QA: FLAN-QA***) | The Chargers would respond with RB LaDainian Tomlinson with a 4-yard TD run. |
| Paraphrase | – President Trump has named Mick Mulvaney to replace John Kelly, the White House chief of staff who left the White House in December." (**Summary: Multi news, Finetuned T5**) | "I am pleased to announce that Mick Mulvaney, Director of the Office of Management & Budget, will be named Acting White House Chief of Staff, replacing General John Kelly, who has served our ... |
| Verbatim | Rachel's son Liam in a house near Glenrothes on 22 March 2014. (***Summary: CNN Daily, FLAN-T5***) | Rachel's son Liam in a house near Glenrothes on 22 March 2014. |
| Verbatim | divided Wednesday during heated arguments over President Obama's health care law, but (***Summary: Multi news, Finetuned T5***) | divided Wednesday during heated arguments over President Obama's health care law, but |
| Verbatim | and liver cirrhosis in dec 2011 modified akt staarted because of cirrhosis i.e (***Dialogue:FLAN-T5, ChatDoctor***) | and liver cirrhosis in dec 2011 modified akt staarted because of cirrhosis i.e |

| Type | Machine-Written Text | Training Text |
|------|----------------------|---------------|
| Verbatim | River Martinez, 10, breaks camp at the Upper Pines Campground in Yosemite National Park, Calif., on Wednesday, July 25, 2018. (*Summary: Multi news, FLAN-T5* ) | River Martinez, 10, breaks camp at the Upper Pines Campground in Yosemite National Park, Calif., on Wednesday, July 25, 2018. |
| Verbatim | A rare blue lobster caught by local lobsterman, Greg Ward, is on display at the Seacoast Science Center in Rye, N.H., on Tuesday, July 18, 2017. (*Summary: Xsum, FLAN-T5*) | A rare blue lobster caught by local lobsterman, Greg Ward, is on display at the Seacoast Science Center in Rye, N.H., on Tuesday, July 18, 2017. |
| Verbatim | Sheffield homered twice and keyed a four-run rally in the ninth inning Thursday night, sending the (*Classification: AG news, FLAN-T5*) | Sheffield homered twice and keyed a four-run rally in the ninth inning Thursday night, sending the |
| Idea | KUALA LUMPUR (Reuters) - Kim Jong Un's half-brother was carrying $100,000 in cash in his backpack at the time of his murder, the officer investigating the case told a police officer" (*Summary: Multi news, Finetuned T5*) | Wan Azirul testified that Kim was carrying $100,000 in cash in his backpack. |
| Idea | Alan Dawson, 64, of Urmston, was convicted of seven counts of indecent assault and one count of rape at Manchester Crown Court. (*Summary: Xsum, FLAN-T5*) | ...is charged with one count of rape and one count of sexual assault. |
| Idea | – Trey Radel, the Florida Rep. who was arrested last month for buying cocaine, is a freshman congressman who has been a big news story for the Washington Post. (*Summary: Multi News, Finetuned T5*) | Post) \n \n Florida Rep. Trey Radel (R-Fla.) was arrested last month for buying cocaine. |
| Idea | Hello, I am 20 years old, height 51, height 51, weight 40kg. (*Dialogue: ChatDoctor, FLAN-T5*) | please help.... hello doctor, m 20 years old, height 5 1 & weight 40kg. |
| Idea | Abdul Aziz believes he was standing right next to a shooter when gunmen opened fire at a parade in new orleans, injuring 19 people. "Everyone around me was right next to a shooter," Abdul Aziz said. (*Summary: CNN Daily, FLAN-T5*) | I was standing, I believe, right next to the shooter. |

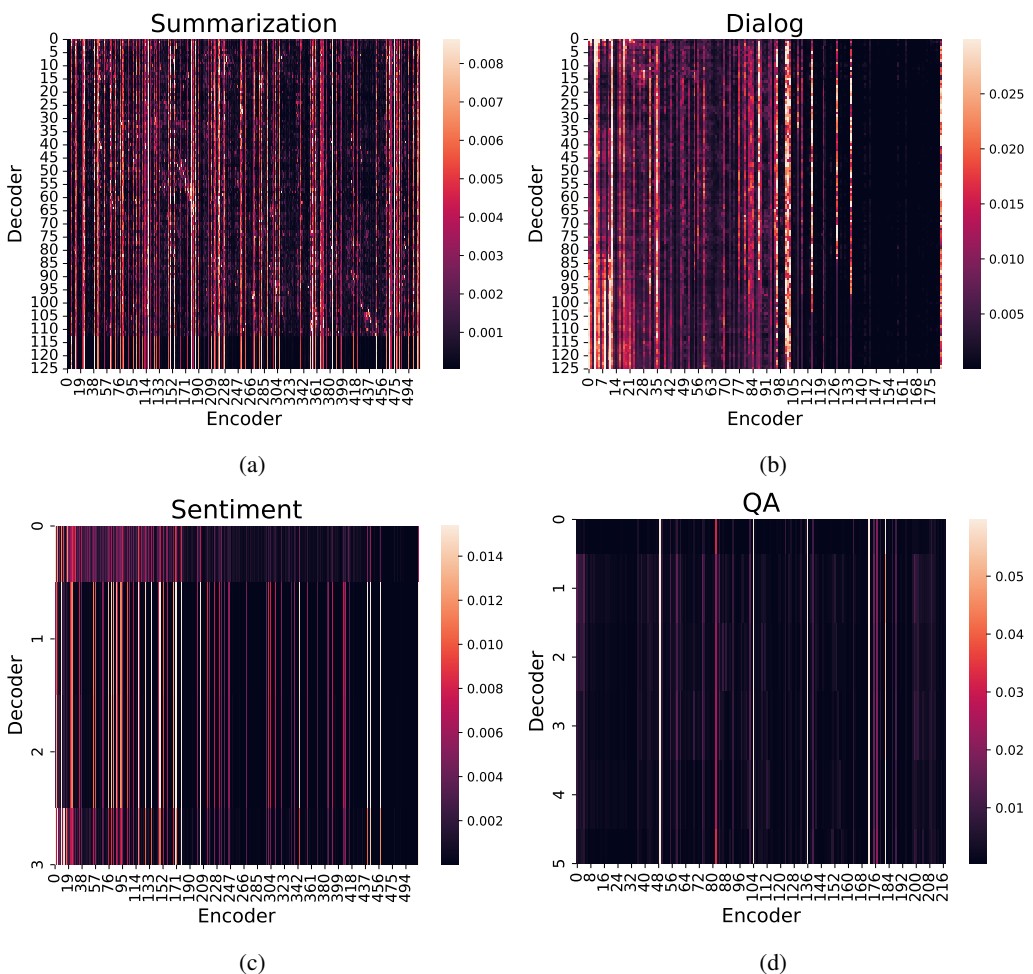

Figure 5: Decode-encoder attention heatmaps on (a) summarization, (b) dialog, (c) sentiment, and (d) QA on T5-base

# F    ATTENTION MAPS

## F.1    ATTENTION MAPS OF T5-BASE WHEN DOING DIFFERENT TASKS

To validate that attention patterns are more intrinsic properties of the tasks themself, we visualize the attention maps of the T5-base model(without fine-tuning) when doing different tasks in Figure 5. Specifically, we use the same instruction and input-output pairs of fine-tuning data as Section 5.1, but just change the model from finetuned-T5 to T5-base. From the Figure we can see that the disparity still exists across different tasks. And for each task, the attention patterns are similar to that of Fine-tuned T5. It further validates that the information needed to complete certain tasks is the intrinsic property of the task.

## F.2    ATTENTION MAPS OF DIFFERENT ENCODER-DECODER LAYERS

Here we visualize the attention maps of different encoder-decoder layers in Figures below. We can clearly observed consistent patterns across various layers of the encoder-decoder attention mechanism, with high memorization tasks showing dense attention and low memorization tasks focusing attention on fewer positions.

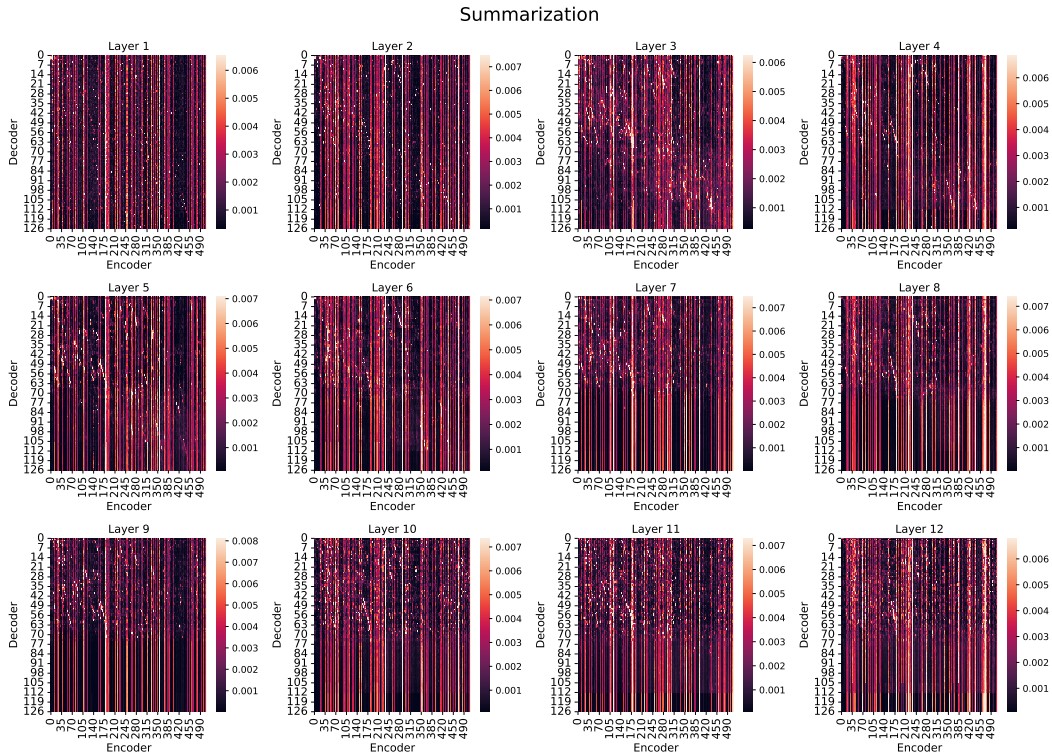

Figure 6: Decode-encoder attention heatmaps on summarization

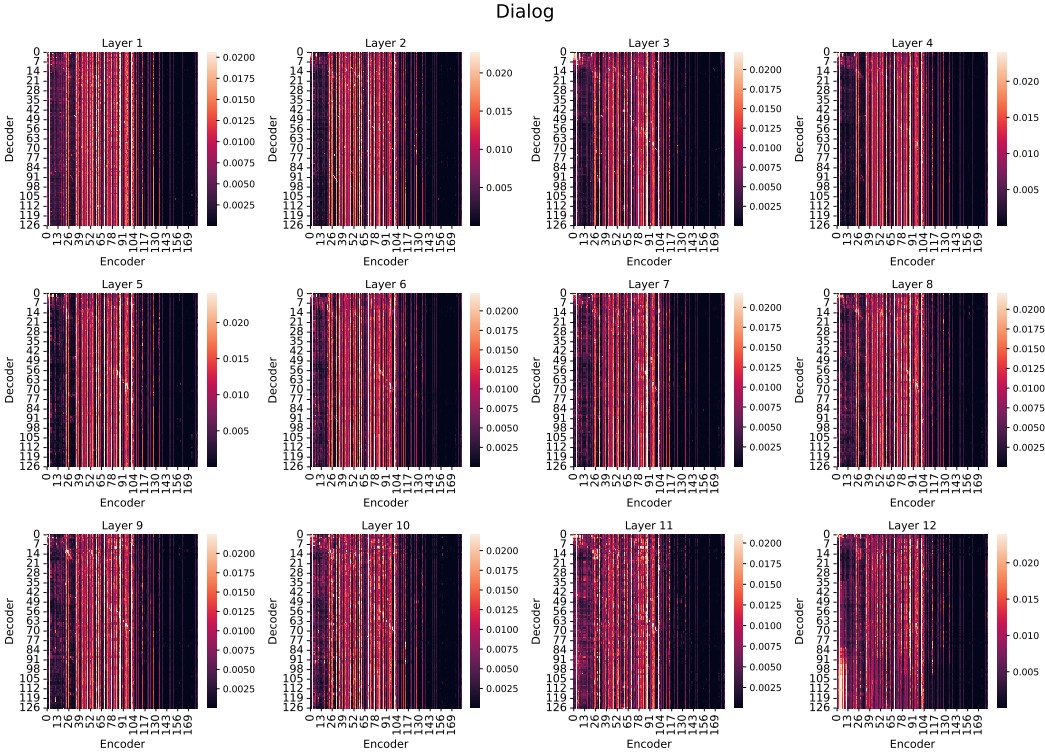

Figure 7: Decode-encoder attention heatmaps on dialog

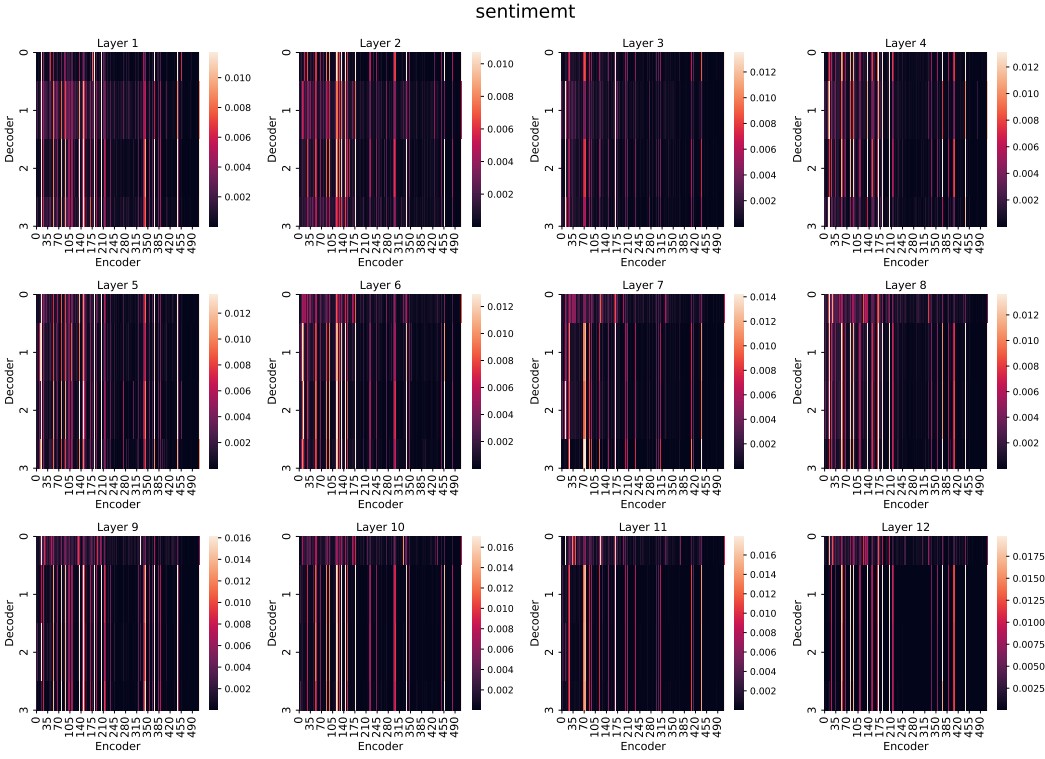

Figure 8: Decode-encoder attention heatmaps on sentiment classification

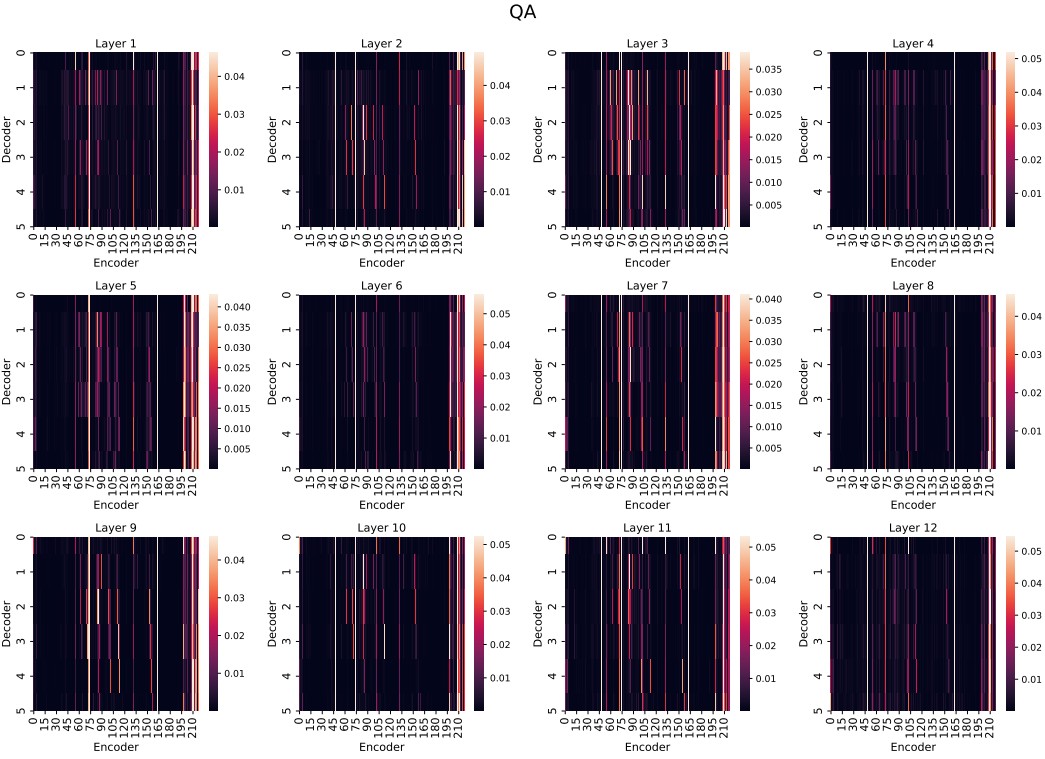

Figure 9: Decode-encoder attention heatmaps on QA

