# OpenReview forum: "Exploring Memorization in Fine-tuned Language Models"
_ICLR.cc/2024/Conference — ICLR 2024 Conference Withdrawn Submission_

### Official Review · Reviewer_wy1y · 2023-11-01

**Soundness:** 1 poor
**Presentation:** 2 fair
**Contribution:** 1 poor
**Rating:** 3
**Confidence:** 4

**Summary:**

The paper investigates to which extent language models (LMs) memorize their fine-tuning datasets, when fine-tuned for different tasks (summarization, medical dialog, QA, translation and sentiment classification). To detect memorization, the authors employ a plagiarism checker tool to detect whether models produce output that can be considered as one of three types of plagiarism (verbatim, paraphrase, idea) of text in the fine-tuning dataset, when prompted with prefixes from the same dataset.
The paper finds that the degree of memorization varies strongly based on the fine-tuning task, with summarization and dialog exhibiting much higher memorization than classification, translation and QA tasks. The paper further shows a correlation between attention distributions and the measured memorization scores, with high memorization corresponding to more uniform attention maps, and low memorization to more concentrated attention maps. Finally, the paper proposes to use multi-task fine-tuning to reduce memorization.

**Strengths:**

- Understanding the memorization behavior of LMs is an important problem with ramifications for privacy and copyright considerations. Since fine-tuning plays a key role in creating usable LMs, the paper tackles an important open problem by investigating how this step impacts memorization.
- Differences in the fine-tuning task can strongly influence model behavior, so investigating how different fine-tuning objectives influence memorization is important.

**Weaknesses:**

- The paper has severe soundness issues, making the key findings rather unreliable.
    1. The paper claims, that memorization varies based on the fine-tuning task, with summarization and dialog tasks exhibiting higher memorization than QA, translation and sentiment classification. It is very likely, however, that this result is confounded by the response lengths required for the different tasks. According to Figure 2, models generate in the order of > 120 output tokens for summarization and dialog tasks, but <= 6 output tokens for sentiment classification and QA tasks. Tasks that require long outputs are much more likely to produce sequences similar to those in the training corpus, just from a numerical perspective, and irrespective of memorization effects. The paper does not control for differences in output lengths of different tasks. Therefore, the results about differences in memorization behavior for different fine-tuning tasks are unreliable without controlling for this confounder.
    2. The results on using attention maps as indicators of memorization appear similarly unreliable, because they also do not control for important potential confounders. Summarization and dialog (presumably high memorization tasks) are shown to have more uniform attention maps, compared to QA and sentiment classification, where attention is more concentrated. However, for the latter two tasks it may be sufficient to pay attention to a few keywords in the input, whereas summarization and dialog presumably require a more comprehensive understanding of the entire text. Therefore, the differences in attention patterns might simply be due to the nature of the different tasks, and not due to different degrees of memorization. For an apples to apples comparison, the authors should compare attention maps of instances with different degrees of memorization on the same task.
    3. In Sections 4.2 and 5.2 the paper uses a model based on sparse coding to theoretically explain task-dependent differences in memorization and in the attention patterns, respectively. However, it is not clear that the sparse coding model is a meaningful approximation of the behavior of the investigated LMs. The sparse coding approach uses a simple linear model, and it is a priori questionable whether such a model is sufficient to approximate large and highly non-linear LMs. Sections 4.2 and 5.2 provide no empirical evidence showing that the sparse coding model is a meaningful approximation of the investigated LMs, so the theoretical arguments made in these sections do not appear to be meaningful.
    4. The paper investigates memorization as a result of fine-tuning models. However, according to Figure 4, verbatim, idea and high-paraphrasation memorization appear to be in the same ballpark for T5-base and fine-tuned models. Therefore, it is not clear how much of the reported memorization is due to fine-tuning and how much is due to pre-training. To account for this, the authors should control for pre-training memorization in the memorization scores of fine-tuned models.
- The paper has several clarity issues which make it difficult to understand some of the results.
    1. The concept of idea memorization/plagiarism and how it is operationalized is not clearly defined. This makes it difficult to interpret the corresponding results.
    2. How is fine-tuning for the different tasks performed? E.g. for sentiment classification, what is the target output ($y$) that the models are fine-tuned to predict? A single word/token, or a short sentence?
    3. In Section 6, how do you use the base/non-fine-tuned models for summarization?
    4. I have difficulty interpreting the memorization examples in Table 11 in Appendix D. Which prompts were used to generate the machine written text?
- Contribution: The memorization detection approach and pipeline in the paper seems to be very similar to that of [A], upon which it builds. Given that the findings in the paper are unreliable (see above), the contribution is very marginal.

[A] Lee et al., Do Language Models Plagiarize?, WWW'23

**Questions:**

1. Is there only one split of each input $x_i$ into prefix $p_i$ and suffix $s_i$? If yes, the memorization behavior of the model might differ, based on the length of prefixes $p_i$, so it might be worth testing memorization for different prefix lengths. If no, the memorization ratio definition seems difficult.
2. What is the rationale for studying attention at the last decoder-encoder attention block in Section 5.1? Do the results look similar for other blocks in the network?

---

> ### Author Response · Authors · 2023-11-18
> **Response to reviewer wy1y**
>
> We are grateful for the insightful feedback and constructive critiques provided during the ICLR 2024 review process. We appreciate the opportunity to address the concerns raised and enhance the clarity and impact of our work. We have thoroughly addressed each point in our rebuttal and hope these improvements justify a reevaluation of our score.
> > **W1：The paper does not control for differences in output lengths of different tasks. Therefore, the results about differences in memorization behavior for different fine-tuning tasks are unreliable without controlling for this confounder.**
>
>
> **Response:** Thanks for your comment. In our experiment, we did not standardize the output length; rather, **we provided the model with a sequence prefix and allowed it to autonomously determine the appropriate output length.** The generation process ceases upon encountering a designated stop symbol. It is accurate, as noted, that certain tasks—such as summarization and dialogue—tend to produce longer outputs in comparison to tasks like sentiment classification or question-answering.
>
>
> To isolate the impact of the output lengh, we **augment the answer of sentiment** **classification task**. Specifically, we change the label "Negative" to "According to the review, this audience gave negative feedback about the movie, indicating a general dissatisfaction with its overall quality and appeal. So the sentiment of this review is negative.", the label "possitive" to "According to the review, the audience shared positive feedback about this movie, highlighting its overall appeal and entertainment value. So the sentiment of this review is positive."(40 words) We call this task "Sentiment-Aug". In this case, the output lengh is comparable to tasks like summarization and dialog. **However, we can find that Sentiment-Aug still represents very low memorization ratio and is much lower than high-memorization tasks.** **This  underscores the nuanced relationship between output length and memorization, which is dependent on the intrinsic demands of the task at hand.**
>
>
> | Task| Dataset| Memorization Ratio | Verbatim | Idea | Paraphrase(p>0.5) | Paraphrase(p<0.5) |
> |-|-|-|-|-|-|-|
> | Summarization  | Multi_news| 22.33%| 4.23%| 0.65%| 6.23%| 11.22%|
> | Dialog| HealthCareMagic | 8.27%| 0.02%| 1.41%| 1.75%| 5.09%|
> | Sentiment| IMDB| 0.80%| 0.04%| 0.30%| 0.17%| 0.29%|
> | Sentiment-Aug | IMDB-Long| 1.17%| 0.03%| 0.34%| 0.23%| 0.66%|
>
> >  **W2: The results on using attention maps as indicators of memorization appear similarly unreliable because they also do not control for important potential confounders. Summarization and dialog (presumably high memorization tasks) are shown to have more uniform attention maps, compared to QA and sentiment classification, where attention is more concentrated. However, for the latter two tasks, it may be sufficient to pay attention to a few keywords in the input, whereas summarization and dialog presumably require a more comprehensive understanding of the entire text. Therefore, the differences in attention patterns might simply be due to the nature of the different tasks, and not due to different degrees of memorization. For an apples-to-apples comparison, the authors should compare attention maps of instances with different degrees of memorization on the same task.**
>
> **Response:**
> * Thank you for your insightful feedback. Indeed, as you have discerned, different tasks exhibit varying dependencies on specific informational elements, a characteristic intrinsic to the nature of the task itself. **Our findings corroborate your observation:**
>     * Tasks that necessitate a comprehensive attention to detail, such as Summarization, demonstrate dense attention maps and a higher memorization ratio.
>     * Conversely, tasks like sentiment classification, which inherently require only a subset of the available information, show markedly lower memorization.
>     * **This suggests that the disparity in memorization across tasks is fundamentally linked to the volume of input information that the task demands for successful completion**.
>
>
> * Furthermore, in response to your suggestion, we conducted a comparative analysis of attention scores and memorization ratios **within the domain of summarization**. By examining T5 models fine-tuned on distinct datasets—Multi-news and CNN_dail—we discovered that the **Multi-news T5 model exhibits both a higher memorization ratio and a greater CAD. Conversely, the CNN_Daily T5 model displays a lower memorization ratio and reduced CAD.** These observations are in alignment with our conclusion, reinforcing the premise that the memorization disparity is attributable to the essential nature of the fine-tuned task.
>
> | Source mode | Task| Dataset| Memorization ratio | CAD|
> |-------------|---------------|------------|---------------------|------|
> | T5-base     | Summarization | CNN_Daily| 5.82%| 2.34 |
> | T5-base     | Summarization | Multi-news | 22.33%| 2.67 |

---

> > ### Author Response · Authors · 2023-11-18
> > **Response to reviewer wy1y**
> >
> > > **W3: In Sections 4.2 and 5.2 the paper uses a model based on sparse coding to theoretically explain task-dependent differences in memorization and in the attention patterns, respectively. However, it is not clear that the sparse coding model is a meaningful approximation of the behavior of the investigated LMs. The sparse coding approach uses a simple linear model, and it is a priori questionable whether such a model is sufficient to approximate large and highly non-linear LMs. Sections 4.2 and 5.2 provide no empirical evidence showing thatthe sparse coding model is a meaningful approximation of the investigated LMs, so the theoretical arguments made in these sections do not appear to be meaningful.**
> > >
> > **Response:**
> > In our paper, our purpose of introducing sparse coding theory is to better illustrate our intuition that the memorization disparity across tasks comes from the natural that different tasks need different amounts of information("True features" in sparse coding theory). We refer to some recent advances in the theory side[1-2] that use sparse coding in the NLP domain and use the theory to explain our intuition. However, the extension of this theory to exactly explain non-linear/huge models is still an open problem in the theory community, which is out of our current scope and we would like to explore it in the future.
> >
> > [1] In-Context Convergence of Transformers.
> >
> > [2]Trained Transformers Learn Linear Models In-Context
> >
> > > **W4: The paper investigates memorization as a result of fine-tuning models. However, according to Figure 4, verbatim, idea and high-paraphrastic memorization appear to be in the same ballpark for T5-base and fine-tuned models. Therefore, it is not clear how much of the reported memorization is due to fine-tuning and how much is due to pre-training. To account for this, the authors should control for pre-training memorization in the memorization scores of fine-tuned models.**
> >
> > **Response:** We compare the finetuned model and base model memorization in tables below:
> > ####  Memorization of T5-base
> >
> > | Task| Dataset| Memorization Ratio | Verbatim | Idea  | Paraphrase (p>0.5) | Paraphrase (p<0.5) |
> > |-|-|-|-|-|-|-|
> > | Summarization| Multi_news| 11.03%| 3.58%| 0.66% | 4.28%| 5.46%|
> > | Dialog| HealthCareMagic| 1.60%| 0.03%| 1.04% | 0.11%| 0.42%|
> > | Sentiment| IMDB| 0.78%| 0.05%| 0.37% | 0.16%| 0.20%|
> > | QA| Squad| 0.09%| 0.02%| 0.00% | 0.01%| 0.05%|
> > | Translation| WMT_16| 0.00%| 0.00%| 0.00% | 0.00%| 0.00%|
> >
> > #### Memorization Difference after fine-tuning
> > | Task| Dataset| Memorization Ratio difference |
> > |---------------|------------------|--------------------------------|
> > | Summarization | Multi_news       | 11.30%|
> > | Dialog| HealthCareMagic  | 6.67%|
> > | Sentiment     | IMDB             | 0.2%                         |
> > | QA            | Squad            | 0.06%                          |
> > | Translation   | WMT_16           | 0.00%                          |
> >
> > As you correctly mentioned, part of the fine-tuning data may also be present in the pre-training data, as T5 also shows some cases of memorization of the fine-tuning data. **However, after fine-tuning, we can clearly see that the fine-tuned model's memorization for summarization and dialogues improves significantly. It is obvious that the model has memorized a large number of fine-tuning data samples during the fine-tuning stage. However, for other tasks such as sentiment classification, the memory ratio only changes, meaning that the model does not remember much information from the fine-tuned data.**
> >
> > > **The concept of idea memorization/plagiarism and how it is operationalized is not clearly defined.**
> >
> >
> > Here we will distinguish 3 types of memorization. First, verbatim memorization means exact copies of words or phrases without transformation. In the cases of paraphrase and idea memorization, the output are not identical to the original text but share similar meanings.  **While paraphrase plagiarism targets sentenceto-sentence transformations, idea plagiarism reads a chunk of the content and condenses its main information into fewer sentences(or vice versa)**.
> > ```
> > Verbatim:
> > Text A: My name is Jack
> > Text B: My name is Jack
> > Paraphrase:
> > Text A: My name is Jack
> > Text B: Jack is my name
> > Idea memorization:
> > Text A： A boy tell me in the class that his name is Jack
> > Text B: A boy is Jack
> > ```
> >
> > In practice of the PAN2014-detection, It starts by identifying closely matched short document fragments (referred to as 'seeds') and then expands these seeds into longer text segments. This is achieved by clustering these fragments based on their separation and employing a 'maxgap' threshold parameter to form coherent clusters. They experimentally find out the most suitable threshold for different plagiarism datasets so that those parameters could be used for the detection of a specific type of memorization. In other words, each memorization case will be counted only once and there will not be overlapping across different categories.

---

> ### Author Response · Authors · 2023-11-18
> **Response to reviewer wy1y**
>
> > **How is fine-tuning for the different tasks performed? E.g. for sentiment classification, what is the target output() that the models are fine-tuned to predict? A single word/token, or a short sentence?**
>
> **Response**: For sentiment classification, we use a single word, either positive or negative as output. During finetuning, we add a prefix for the input text: “Please classify the sentiment of the following paragraph: ” .For extractive QA, the output data which are the corresponding answer of the questions from the input data, is usually several words or a short sentence. We finetune this task in a sequence-to-sequence manner thus use the answer term as output. For other tasks like summarization, translation, and dialogue, we directly use the sentences presented in the original dataset as output. In our experiment, we consider all tasks as the format of generation tasks.(Like instruction tuning)
>
> > **In Section 6, how do you use the base/non-fine-tuned models for summarization?**
>
> **Response**:
> In our experiment, the test procedure is the same for both base models and finetuned models. It means that we input the prefix $p$ of finetuning data  to models and compare the $f(p)$ with suffix $s$. For base model, we still input the finetuning data prefix for test, even if the model is not finetuned on the dataset.(So if the model remember the data means the data also presents in the pre-training set)
> >
> > **I have difficulty interpreting the memorization examples in Table 11 in Appendix D. Which prompts were used to generate the machine-written text?**
>
> **Response:** In our experiment to examine memorization, we don't have a task-specific prompt during the test time. The Machine-written text is the output of the model when we input the prefix of the original input. We follow the memorization inference setting from previous work[3].
>
> [3] Quantifying Memorization Across Neural Language Models
>
> > **Q1: Is there only one split of each input into prefixes and suffixes? If yes, the memorization behavior of the model might differ, based on the length of prefixes, so it might be worth testing memorization for different prefix lengths. If not, the memorization ratio definition seems difficult**.
>
> **Response:**
> Thanks for your question, we conduct ablation studies on Dialog,Summarization, and Sentiment Classification with differnt prefixs k. and reported the result on Appendix B.3. And the results show that:
> * **The length of prefix tokens can affect memorization**.
>     * The length of prefix tokens does indeed impact memorization. Specifically, for summarization and Dialog tasks, the memorization ratio generally increases with the length of the prefix. This finding aligns with previous research on pretrained memorization[^2]. However, for sentiment classification, changing the prefix does not result in significant changes, and increasing the prefix length does not necessarily lead to an increase in the memorization ratio.
> * **The task disparity still exists when using different prefix lengths.**
>     * Furthermore, it is worth noting that despite the influence of different prefixes on memorization, there still exists a noticeable disparity in memorization across tasks. Therefore, our conclusion remains even using different prefixes.
>
>
> | Task| Dataset| Prefix length | Memorization Ratio | Verbatim | Idea  | Paraphrase (p>0.5) | Paraphrase (p<0.5) |
> |---------------|------------|---------------|---------------------|-----------|--------|----------------------|---------------------|
> | Summarization | Multi_news | 10| 12.25%| 1.74%| 2.85%  | 0.88%| 6.78%|
> | Summarization | Multi_news | 30| 20.68%| 7.07%| 1.41%  | 3.05%| 9.15%|
> | Summarization | Multi_news | 50| 22.33%| 4.23%| 0.65%| 6.23%| 11.22%|
> | Summarization | Multi_news | 100| 29.66%| 10.61%| 0.79%  | 4.27%| 13.99%|
>
>
> | Task   | Dataset| Prefix length | Memorization Ratio | Verbatim | Idea| Paraphrase (p>0.5) | Paraphrase (p<0.5) |
> |--------|----------------|---------------|---------------------|----------|-------|---------------------|--------------------|
> | Dialog | HealthCareMagic | 10| 6.28%| 0.03%| 1.94% | 0.85%| 3.46%|
> | Dialog | HealthCareMagic | 30| 7.76%| 0.04%| 1.28% | 1.72%| 4.72%|
> | Dialog | HealthCareMagic | 50| 8.27%| 0.02%| 1.41% | 1.75%| 5.09%|
>
>
> | Task| Dataset | Prefix length | Memorization Ratio | Verbatim | Idea  | Paraphrase (p>0.5) | Paraphrase (p<0.5) |
> |-----------|---------|---------------|---------------------|----------|-------|---------------------|--------------------|
> | Sentiment | IMDB    | 10| 1.37%| 0.00%| 1.12% | 0.06%| 0.19%|
> | Sentiment | IMDB    | 30| 1.18%| 0.01%| 0.51% | 0.15%| 0.51%|
> | Sentiment | IMDB    | 50| 0.80%| 0.04%| 0.30% | 0.17%| 0.29%|
> | Sentiment | IMDB    | 100| 1.39%| 0.05%| 0.23% | 0.33%| 0.78%|

---

> > ### Author Response · Authors · 2023-11-18
> > **Response to reviewer wy1y**
> >
> > > **Q2: What is the rationale for studying attention at the last decoder-encoder attention block in Section 5.1? Do the results look similar for other blocks in the network?**
> > >
> > **Response:**
> > In our T5 architecture study, we examined three types of attention layers: encoder, decoder, and encoder-decoder attention layers. Our focus on encoder-decoder attention layers stems from their ability to capture the attention distribution across input features for each output, unlike encoder and decoder attention layers, which track internal attention within inputs and outputs respectively.
> >
> > This choice is based on the hypothesis that memorization behavior differences are linked to the specific information requirements for task completion. Encoder-decoder attention scores are thus more aligned with our research objectives.
> >
> > **We observed consistent patterns across various layers of the encoder-decoder attention mechanism**, with high memorization tasks showing dense attention and low memorization tasks focusing attention on fewer positions. Due to this consistency, we report primarily on the final layer, closest to the output, for clarity and relevance. Detailed attention maps across different layers are now available in the Appendix-F.2 for further reference.
> >
> > > **Contribution: The memorization detection approach and pipeline in the paper seems to be very similar to that of [A], upon which it builds. Given that the findings in the paper are unreliable (see above), the contribution is very marginal**
> >
> > **Response:** Thank you for highlighting our methodological approach. We would like to emphasize that our primary contribution lies in elucidating the memorization characteristics of fine-tuned Language Models (LMs), rather than developing a new plagiarism detection pipeline. Our focus is on uncovering the nuanced memorization tendencies that emerge during the fine-tuning phase, a stage critical for two main reasons:
> >
> > * **Sensitivity of Data:** The fine-tuning process often involves handling more sensitive information, making an understanding of memorization patterns particularly pertinent.
> >
> > * **Diversity in Tasks and Objectives:** The variety of tasks and objectives inherent to the fine-tuning stage leads to a wide range of memorization behaviors. Our research systematically reveals these behaviors across different tasks, offering crucial insights and understanding of the disparities in memorization.
> >
> > These findings are not only theoretically significant but also carry substantial practical implications for privacy considerations in model fine-tuning. Our work aids model builders in making informed decisions when fine-tuning models on specific tasks, particularly from a privacy perspective. So we believed that our research is positive to the field.

---

> ### Author Response · Authors · 2023-11-20
> **A friendly reminder**
>
> We are grateful for the useful comments provided by you. We hope that our answers have addressed your concerns. If you have any further concerns, please let us know. We are looking forward to hearing from you.

---

> ### Author Response · Authors · 2023-11-21
> **A friendly reminder**
>
> We appreciate your reviews. We hope that our responses have adequately addressed your concerns. As the deadline for open discussion nears, we kindly remind you to share any additional feedback you may have. We are keen to engage in further discussion.

---

> ### Comment · Reviewer_wy1y · 2023-11-22
>
> Thank you for the extensive response and for responding to all of my questions and concerns!
> Your answers address a lot of the clarity issues I had with the work, and it's good to see the additional results you provide.
> My main objections regarding soundness remain, however, so I will maintain my score.
>
> Here are my thoughts on the response:
> - W1: Sentiment-Aug task does not really address the problem of controlling for output length. The additional output tokens are just due to a fixed template that the model regurgitates, whereas in summarization and dialog the model produces potentially completely different token sequences depending on the input. Overall I'm not convinced that the notion of memorization based on whether sequences from the finetuning data are repeated is the right one for a task like sentiment classification. A more appropriate notion would seem to be based on detecting whether the assigned label strongly depends on certain substrings that also appeared in the finetuning data. It may be worth taking a look at work that proposed label memorization metrics for classification tasks, although it has been done in the vision domain and would have to be adapted to the NLP setting: https://arxiv.org/abs/1906.05271. I would consider developing separate approaches to memorization detection in tasks that require text generation vs classification or localization, since both the output formats as well as what constitutes memorization at a conceptual level differ significantly.
> - W2: Thank you for your clarification. The results shown are a good first step, but I think it is necessary to investigate whether the attention-memorization trend holds more broadly to infer a statistically significant correlation.
> - W4: It's good to see these additional numbers. I think the post-finetuning scores should be normalized by the pre-finetuning scores to make sure the measure is reporting the right results.
> - Q1: Thank you for running the additional analysis. It is interesting to see that different prefix lengths recall memorized text to different extents.

---

> > ### Author Response · Authors · 2023-11-22
> >
> > Thank you for your valuable reply and follow-up questions. We appreciate your positive feedback on this work. Below we provide a detailed explanation of your question. We hope it can solve your concerns and merit a better score.
> > > W1: Sentiment-Aug task does not really address the problem of controlling for output length. The additional output tokens are just due to a fixed template that the model regurgitates, whereas in summarization and dialog, the model produces potentially completely different token sequences depending on the input. Overall I'm not convinced that the notion of memorization based on whether sequences from the finetuning data are repeated is the right one for a task like sentiment classification. A more appropriate notion would seem to be based on detecting whether the assigned label strongly depends on certain substrings that also appeared in the finetuning data......
> >
> > **Response:** Thanks for your follow-up question. The first thing we need to clarify is that  **LLMs actually treat all tasks as generative tasks**. In our fine-tuning setup, we treat all tasks as generation tasks and unify the format as instruction tuning. In our test setup, we also did not use task prompts to ask the model to perform a specific task, but instead simply fed the model a prefix and then saw if the model produced similar output as the suffix. **Since all tasks can be considered generative tasks, we believe it is reasonable to use a unified memory definition for all tasks of generative models.**
> >
> > Thanks for your suggestion on the label memorization. It  adeptly identifies the correlation between inputs and outputs, a crucial aspect of understanding memorization. In this work, we aim to explore the extent to which a model might memorize and reproduce specific inputs (denoted as 'x'). This focus stems from heightened concerns regarding privacy implications. For instance, in sentiment classification, the inadvertent revelation of labels such as 'positive' or 'negative' may pose limited risk. However, the exposure of 'x', which could be a personal message, presents a significantly greater privacy risk. Therefore our focus is quite different and we would like to explore it as one future work.
> >
> >
> >
> > > W2: Thank you for your clarification. The results shown are a good first step, but I think it is necessary to investigate whether the attention-memorization trend holds more broadly to infer a statistically significant correlation.
> >
> > **Response:** Thanks for your positive feedback. In our experiment, we have included serveral most representive tasks and consistently found important correlations between attention patterns and memorization. We believe our insights  can generalize to a broader set of tasks.
> >
> > > W4: It's good to see these additional numbers. I think the post-finetuning scores should be normalized by the pre-finetuning scores to make sure the measure is reporting the right results.
> >
> > **Response:** Thanks for your reply. Here we want to clarify that attention distribution is an intrisic property of the task.
> > * **Attention scores inherently represent the unique characteristics of each task:** Diverse tasks have varying requirements regarding the extent to which they necessitate focus on the entire set of presented information. These requirements are aptly mirrored in the attention heatmap patterns. **We visualize the attention heatmap of the T5 base (non-finetuned) when we use the non-finetuned model to perform different tasks. We were surprised to find, in Appendix F.1 and Figure 5, that the patterns still differed significantly across tasks (high memory - intensive attention).** It further verifies that attention distribution is an intrinsic property of the task itself. In other words, taking summarization as an example, no matter what model we use (even the human brain) to complete the task, we need to pay attention to the details of the input to complete the task. If we train the model on this "high memory" task, the model tends to remember more.
> > * **Practical Implications:** One use case might be: before fine-tuning a model for a specific task, the model builder could first test attention patterns for that task by inferring other models (e.g., pre-trained models). Based on the attention pattern, the model builder can know whether the training data will be easily remembered if we fine-tune it for this task.
> >
> > Attention maps are intrinsic properties of the task. Even if we use T5-base for summarization tasks, the scores are still high. **We would like to clarify that the reason for high memory is because the task itself requires intensive attention, not because the attention map changes after fine-tuning (for example, it was originally sparse and became dense after fine-tuning).** In this case, it is not necessary to normalize the scores. (In fact, we can even use pretrained models to determine whether a task is a high-memory task.)

---

### Official Review · Reviewer_BVwv · 2023-11-06

**Soundness:** 2 fair
**Presentation:** 2 fair
**Contribution:** 2 fair
**Rating:** 3
**Confidence:** 5

**Summary:**

1. The paper investigates the phenomenon of memorization within fine-tuned language models, specifically focusing on pre-trained T5 models applied to various downstream tasks, such as summarization, dialogue, QA, translation, and sentiment analysis.
2. The primary goal of the study is to comprehend the variations in memorization across these diverse tasks and determine whether fine-tuned models exhibit task-specific memorization patterns.
3. The paper introduces a categorization of memorization into three types: verbatim (exact memorization), paraphrase (memorization of alternative expressions), and idea (conceptual memorization).
4. Additionally, the authors propose a novel method for estimating memorization using attention scores and demonstrate the possibility of reducing memorization through multitask fine-tuning.

**Strengths:**

1. I like the idea of simplifying the concept of memorization by linking it to the extent of information required for a given task, which aligns with the sparse coding model.
2. The paper offers a systematic analysis that spans various tasks, providing valuable insights into memorization patterns across these tasks and comparing memorization across different language models.
3. The examination of attention scores and the presentation of encoded attention maps explains the initial claim well.
4. The observation that multitask fine-tuning can lead to reduced memorization highlights the practical implications of the study for model training and application.

**Weaknesses:**

1. Need for a more rigorous analysis of task specificity: The authors should consider potential confounding factors, especially the influence of the pre-training data used for the model, as this could impact memorization ratios. For instance, in the final section, the authors show that the T5 base model also shows a very high memorization ratio on the summarization task of multi-news which is 110 memorization ratio and the fine-tuned T5 model only goes to 222 which is twice more. However, the entire analysis of the paper before that was talking about how the summarization task requires more memorization because fine-tuning happened on that particular data set. I believe that a proper analysis of what was there in the pre-training data of a model and whether it confounds the memorization ratios is extremely important for such analysis and any conclusions about the task specificity or the nature of the amount of information required for the task. In fact, in the T5 base model that the authors chose, the entire C4 data set which it was trained on is totally public and the author should make such an analysis so that we can understand what is actually responsible.

2. The rationale behind introducing attention discrimination as a metric is unclear. If it merely correlates with existing, computationally cheaper metrics, it may not be necessary to introduce an additional metric without a compelling justification.

3. The experiment involving Flan T5, where the memorization ratio decreases after fine-tuning, lacks a clear explanation. The authors should provide a rationale for this unexpected result or revisit their analysis.

4. To draw meaningful conclusions, it is essential for the authors to compare the memorization scores of the T5 base model before and after fine-tuning.

5. The authors' choice of focusing on the last layers of encoder-decoder attention (as seen in Figure 2) appears somewhat arbitrary. An explanation for this choice and its potential impact on the correlation between attention scores should be provided.
6. Expanding the analysis presented in Figure 4 to include tasks beyond summarization would enhance the comprehensiveness of the paper's findings.
7. The concept of discriminating between different types of memorization: I believe idea memorization which encapsulates the fundamental sense of the initial information, should be a superset of all other forms of memorization. So, I do not understand this whole concept of categorizing memorization and what are we trying to achieve with this.
8. Please use % and not %. the latter is very non standard for ML papers.

**Questions:**

See weaknesses

---

> ### Author Response · Authors · 2023-11-18
> **Response to Reviewer BVwv**
>
> Thank you for your valuable feedback and constructive comments in the review of our submission for ICLR 2024. We appreciate the opportunity to address the concerns raised and enhance the clarity and impact of our work. We believe that the revisions and additional experiments we propose, as detailed in this rebuttal, will significantly strengthen our paper. Therefore, we respectfully request the reviewers to reconsider the scoring in light of these improvements and our responses to the feedback provided.
> > Q1: Need for a more rigorous analysis of task specificity: The authors should consider potential confounding factors,**especially the influence of the pre-training data** used for the model, as this could impact memorization ratios.
>
> > Q4: To draw meaningful conclusions, the authors need to compare the memorization scores of the T5 base model before and after fine-tuning
>
> **Response:** Thanks for your valuable questions. we add experiment on Appendix B.1 to compare the memorization difference between T5-base and fine-tuned T5 to verify our conclusion.
>
> ####  Memorization of T5-base
>
> | Task          | Dataset        | Memorization Ratio | Verbatim | Idea  | Paraphrase (p>0.5) | Paraphrase (p<0.5) |
> |---------------|----------------|---------------------|----------|-------|---------------------|--------------------|
> | Summarization | Multi_news     | 11.03%              | 3.58%    | 0.66% | 4.28%              | 5.46%              |
> | Dialog        | HealthCareMagic| 1.60%               | 0.03%    | 1.04% | 0.11%              | 0.42%              |
> | Sentiment     | IMDB           | 0.78%               | 0.05%    | 0.37% | 0.16%              | 0.20%              |
> | QA            | Squad          | 0.09%               | 0.02%    | 0.00% | 0.01%              | 0.05%              |
> | Translation   | WMT_16         | 0.00%               | 0.00%    | 0.00% | 0.00%              | 0.00%              |
>
> #### Memorization Difference after fine-tuning
> | Task          | Dataset          | Memorization Ratio difference |
> |---------------|------------------|--------------------------------|
> | Summarization | Multi_news       | 11.30%                         |
> | Dialog        | HealthCareMagic  | 6.67%                          |
> | Sentiment     | IMDB             | 0.2%                         |
> | QA            | Squad            | 0.06%                          |
> | Translation   | WMT_16           | 0.00%                          |
>
> As you correctly mentioned, some of the fine-tuning data may also be present in the pre-training data, as T5 also shows some cases of memorization of the fine-tuning data. However, after fine-tuning, we can clearly see that the fine-tuned model's memorization for summarization and dialogues improves significantly. It is obvious that the model has memorized a large number of fine-tuning data samples during the fine-tuning stage. However, for other tasks such as sentiment classification, the memorization ratio does not change much, meaning that the model does not remember much information from the fine-tuned data.
> > Q2:  **The rationale behind introducing attention discrimination as a metric is unclear**
>
> The reasons we introduce attention scores as an indicator are:
> * **Intrinsic Task Properties:**
>     * Attention scores inherently represent the unique characteristics of each task. Diverse tasks have varying requirements regarding the extent to which they necessitate focus on the entire set of presented information. These requirements are aptly mirrored in the attention heatmap patterns.
>     * We visualize the attention heatmap of the T5 base (non-finetuned) when we use the non-finetuned model to perform different tasks. We were surprised to find, in Appendix F.1 and Figure 5, that the patterns still differed significantly across tasks (high memory-intensive attention). It further verifies that attention distribution is an intrinsic property of the task itself. In other words, taking summary as an example, no matter what model we use (even the human brain) to complete the task, we need to pay attention to the details of the input to complete the task. If we train the model on this "high memory" task, the model tends to remember more.
>
> *  **Robustness of Attention Scores**: Compared to memorization ratios, attention scores are less susceptible to external variables such as model type, training duration, and hyperparameters. This robustness makes them a more reliable indicator of a task's intrinsic characteristics.
> *  **Practical Implications**: One use case might be: before fine-tuning a model for a specific task, the model builder could first test attention patterns for that task by inferring other models (e.g., pre-trained models). Based on the attention pattern, the model builder can know whether the training data will be easily remembered if we fine-tune it for this task.

---

> ### Author Response · Authors · 2023-11-18
> **Response to Reviewer BVwv**
>
> > Q3:**The experiment involving Flan T5, where the memorization ratio decreases after fine-tuning, lacks a clear explanation. The authors should provide a rationale for this unexpected result or revisit their analysis.**
> >
> In our study, we conducted memorization tests on the instruction-tuning set of Flan-T5, which encompasses a variety of tasks. Our findings reveal a notable decrease in the memorization rate when compared to models fine-tuned on single tasks. A plausible explanation for this observation is the enhanced generalization capability of the model in a multi-task setting. Exposure to a broader array of tasks and data appears to facilitate this improved generalization[1]. Consequently, the model's ability to execute tasks is not solely dependent on memorization but also on its refined generalization skills. This explanation has been incorporated into our manuscript to elucidate the observed decrease in memorization rates. Nevertheless, we acknowledge that the underlying causes of this intriguing phenomenon are likely complex and multifaceted and requires further investigation.
>
> > Q5. **The authors' choice of focusing on the last layers of encoder-decoder attention (as seen in Figure 2) appears somewhat arbitrary. An explanation for this choice and its potential impact on the correlation between attention scores should be provided.**
> >
>
> In our T5 architecture study, we examined three types of attention layers: encoder, decoder, and encoder-decoder attention layers. Our focus on encoder-decoder attention layers stems from their ability to capture the attention distribution across input features for each output, unlike encoder and decoder attention layers, which track internal attention within inputs and outputs, respectively.
>
> This choice is based on the hypothesis that memorization behavior differences are linked to the specific information requirements for task completion. Encoder-decoder attention scores are thus more aligned with our research objectives.
>
> **We observed consistent patterns across various layers of the encoder-decoder attention mechanism**, with high memorization tasks showing dense attention and low memorization tasks focusing attention on fewer positions.(And we add the attention maps of other encoder-decoder layers in Appendix-F.2) Due to this consistency, we report primarily on the final layer, closest to the output, for clarity and relevance. Detailed attention maps across different layers are now available in the Appendix-F.2 for further reference.
>
> > Q7. **The concept of discriminating between different types of memorization: I believe idea memorization which encapsulates the fundamental sense of the initial information, should be a superset of all other forms of memorization. So, I do not understand this whole concept of categorizing memorization and what are we trying to achieve with this**.
> >
> Thank you for the question. We  want to clarify that these three types of memorization cases are non-overlapping and idea memorization is not a superset of others. Here we will distinguish 3 types of memorization. First, verbatim memorization means exact copies of words or phrases without transformation. In the cases of paraphrase and idea memorization, the output are not identical to the original text but share similar meanings.  **While paraphrase plagiarism targets sentenceto-sentence transformations, idea plagiarism reads a chunk of the content and condenses its main information into fewer sentences(or vice versa)**.  It is crucial to report and distinguish these types of memorization, as each poses unique implications for privacy concerns. By categorizing and examining these memorization behaviors, we aim to deepen the understanding of language models' capacity for retaining and reproducing information.
> ```
> Examples:
> Verbatim:
> Text A: My name is Jack
> Text B: My name is Jack
>
> Paraphrase:
> Text A: My name is Jack
> Text B: Jack is my name
>
> Idea plagirism:
> Text A： A boy tell me in the class that his name is Jack
> Text B: A boy is Jack
> ```
>
> In practice of the PAN2014-detection, It starts by identifying closely matched short document fragments (referred to as 'seeds') and then expands these seeds into longer text segments. This is achieved by clustering these fragments based on their separation and employing a 'maxgap' threshold parameter to form coherent clusters.They experimentally find out the most suitable threshold for different plagiarism datasets so that those parameters could be used for detection of a specific type of memorization. In another word, each memorization cases will be count only once and there will not be overlapping across different catagories.
>
>
>
>
> > Q8. **Please use % and not %. the latter is very non-standard for ML papers.**
>
> Thanks for your comment. We have changed the symbol in the revision.
>
> [1]: Scaling Instruction-Finetuned Language Models

---

> ### Author Response · Authors · 2023-11-20
> **A friendly reminder**
>
> We are grateful for the useful comments provided by you. We hope that our answers have addressed your concerns. If you have any further concerns, please let us know. We are looking forward to hearing from you.

---

> ### Author Response · Authors · 2023-11-21
> **A friendly reminder**
>
> We appreciate your reviews. We hope that our responses have adequately addressed your concerns. As the deadline for open discussion nears, we kindly remind you to share any additional feedback you may have. We are keen to engage in further discussion.

---

### Official Review · Reviewer_gHsj · 2023-11-10

**Soundness:** 3 good
**Presentation:** 3 good
**Contribution:** 3 good
**Rating:** 6
**Confidence:** 3

**Summary:**

The paper studies memorization in fine-tuned language models. The authors observe a higher degree of memorization for models trained on summarization tasks, compared to simpler tasks like sentiment classification. Furthermore, they relate entropy in attention patterns to the degree of memorization in fine-tuned language models. Finally, they show that multi-task training reduces some memorization. They conclude their observations with a short theory on sparse coding models.

**Strengths:**

The main strength of the paper lies in its simplistic approach to studying memorization in fine-tuned language models. The authors take insights from existing memorization works on pre-trained models to conduct their analysis. The differences between different tasks showcase different mechanisms that models employ for each of them. Furthermore, the short theory on the sparse coding model helps formalize a reader's intuition for the observations. Overall, I believe the paper is a positive contribution to the community.

**Weaknesses:**

I have a couple of questions about the experimental setup and the observations that the authors draw from their results.

(a) What does x% memorization mean in the experiments? It would be great to demonstrate perfect and no memorization baselines to get a sense of the numbers.

(b) How do the authors measure Idea memorization?  Furthermore, how do they differentiate Idea memorization from summarization (which is the task of a summarization-tuned model)?

(c) How much do hyperparameters during fine-tuning affect the memorization results? Does a lower learning rate and longer training time result in more memorization?

(d) "k", the length of the prefix tokens, was fixed for all the experiments. How much do the observations vary with varying "k"?

(e) What is the decoding algorithm used for generation? Is it greedy decoding? If so, will the observations change with a more nuanced decoding algorithm, like nucleus sampling? This might allow the model to change its generation output, which will reduce the frequency of the generation of perfectly memorized solutions.

Overall, I believe the paper is a positive contribution to the community. I am happy to interact further with the authors during the rebuttal period.

**Questions:**

Please check my questions in the previous section.

---

> ### Author Response · Authors · 2023-11-18
> **Response to reviewer gHsj**
>
> Thank you for your valuable feedback and constructive comments in the review of our submission for ICLR 2024. We appreciate your positive feedback and the opportunity to address the concerns raised and enhance the clarity and impact of our work. We believe that the revisions and additional experiments we propose(as detailed in this rebuttal) will significantly strengthen our paper . We hope this rebuttal demonstrates the advances and merits a higher score.
>
> > **(a) What does x% memorization mean in the experiments? It would be great to demonstrate perfect and no memorization baselines to get a sense of the numbers.**
> >
> **Response:** In our experiment, we input prefix $p_{i}$ to the model and compare $f(p_{i})$ with the all suffixes $\{s\}$ to identify whether it constitutes memorization. A memorization rate of x\% indicates that an equivalent proportion of the prefixes induces the model to produce memorized content. A rate of 0\% implies no memorized content is produced, whereas a rate of 100\% indicates that all prefixes lead to memorized output. This metric is essential for assessing the model's propensity to replicate known information.
>
> > **(b) How do the authors measure Idea memorization? How do they differentiate Idea memorization from summarization?**
>
> **Response:** In our research, although summarization and idea memorization share similarities, they actually refer to different concepts. Summarization is a task that the model conducts. The user inputs x(e.g. news), and the model should output the summarization of the input(e.g. The title). However, the idea memorization in our paper means a different thing. We input the part of information of x(x_prefix) to the model, if the model outputs some key information of the remaining part of x(x_suffix), we regard it as Idea memorization. E.g. We input the beginning of the news and the model tells us information of the remaining part of the news.
>
> > **(d) How much do the observations vary with varying "k"?**
>
> **Response:** Thanks for your question, we conducted ablation studies on dialog, summarization, and sentiment classification with different prefixes k and reported the result on Appendix B.3. And the results show that:
> * **The length of prefix tokens can affect memorization**.
>     * The length of prefix tokens does indeed impact memorization. Specifically, for summarization and Dialog tasks, the memorization ratio generally increases with the length of the prefix. This finding aligns with previous research on pre-trained memorization[1]. However, for sentiment classification, changing the prefix does not result in significant changes, and increasing the prefix length does not necessarily lead to an increase in the memorization ratio.
>
> [1] Quantifying Memorization Across Neural Language Models
> * **The task disparity still exists when using different prefixes.**
>     * Furthermore, it is worth noting that despite the influence of different prefixes on memorization, there still exists a noticeable disparity in memorization across tasks. Therefore, our conclusion remains even using different prefixes.
>
>
> | Task          | Dataset    | Prefix length | Memorization Ratio | Verbatim | Idea  | Paraphrase (p>0.5) | Paraphrase (p<0.5) |
> |---------------|------------|---------------|---------------------|-----------|--------|----------------------|---------------------|
> | Summarization | Multi_news | 10| 12.25%| 1.74%| 2.85%| 0.88%| 6.78%|
> | Summarization | Multi_news | 30| 20.68%| 7.07%| 1.41%| 3.05%| 9.15%|
> | Summarization | Multi_news | 50| 22.33%| 4.23%| 0.65%  | 6.23%| 11.22%|
> | Summarization | Multi_news | 100| 29.66%| 10.61%| 0.79%| 4.27%| 13.99%|
>
>
> | Task   | Dataset        | Prefix length | Memorization Ratio | Verbatim | Idea  | Paraphrase (p>0.5) | Paraphrase (p<0.5) |
> |--------|----------------|---------------|---------------------|----------|-------|---------------------|--------------------|
> | Dialog | HealthCareMagic | 10| 6.28%| 0.03%    | 1.94% | 0.85%              | 3.46%              |
> | Dialog | HealthCareMagic | 30| 7.76%| 0.04%    | 1.28% | 1.72%| 4.72%|
> | Dialog | HealthCareMagic | 50| 8.27%              | 0.02%    | 1.41% | 1.75%              | 5.09%              |
>
>
> | Task      | Dataset | Prefix length | Memorization Ratio | Verbatim | Idea  | Paraphrase (p>0.5) | Paraphrase (p<0.5) |
> |-----------|---------|---------------|---------------------|----------|-------|---------------------|--------------------|
> | Sentiment | IMDB    | 10| 1.37%| 0.00%    | 1.12% | 0.06%| 0.19%|
> | Sentiment | IMDB    | 30| 1.18%| 0.01%    | 0.51% | 0.15%              | 0.51%              |
> | Sentiment | IMDB    | 50            | 0.80%              | 0.04%    | 0.30% | 0.17%              | 0.29%              |
> | Sentiment | IMDB    | 100           | 1.39%              | 0.05%    | 0.23% | 0.33%              | 0.78%              |

---

> > ### Author Response · Authors · 2023-11-18
> > **Response to reviewer gHsj**
> >
> > > **(e) What is the decoding algorithm used for generation? If so, will the observations change with a more nuanced decoding algorithm.**
> >
> > **Response:** Thanks for your question. In our main results, we all use gready sampling for each task. We add ablation studies for different decoding methods on Appendix B.2. Following the experiment setting of previous paper[^1], we use top-K(K=40) sampling, top-p(0.8<p<1) sampling, and change the temprature to 1(T=1). And the results show that:
> > * For high-memory tasks such as summarization and dialogue,  sampling can reduce the memory rate and change the category distribution of memory samples.
> > *  For low-memory tasks such as emotion classification, sampling does not reduce the memory rate (even increases it in some way), and the changes are not very profound.
> > *  Regardless the decoding methodology employed, **a pronounced disparity in memorization across different tasks persists**. This suggests an inherent task-specific propensity towards memorization that is not substantially mitigated by variations in sampling techniques.
> >
> > | Task           | Dataset    | Decoding Method    | Memorization Ratio | Verbatim | Idea  | Paraphrase(p>0.5) | Paraphrase(p<0.5) |
> > |----------------|------------|--------------------|--------------------|----------|-------|--------------------|-------------------|
> > | Summarization  | Multi_news | Top-K K=40         | 10.80%             | 2.54%    | 0.34% | 1.94%              | 5.98%             |
> > | Summarization  | Multi_news | Top-p 0.8<p<1      | 13.57%             | 4.07%    | 0.54% | 2.26%              | 6.70%             |
> > | Summarization  | Multi_news | temp T=1           | 5.82%              | 1.28%    | 0.23% | 0.83%              | 3.48%             |
> > | Summarization  | Multi_news | Greedy             | 22.33%             | 4.23%    | 0.65% | 6.23%              | 11.22%            |
> >
> >
> > | Task   | Dataset        | Decoding Method  | Memorization Ratio | Verbatim | Idea  | Paraphrase (p>0.5) | Paraphrase (p<0.5) |
> > |--------|----------------|------------------|---------------------|----------|-------|---------------------|--------------------|
> > | Dialog | HealthCareMagic | Top-K K=40      | 5.76%              | 0.05%    | 0.38% | 0.90%              | 4.43%              |
> > | Dialog | HealthCareMagic | Top-p 0.8<p<1   | 7.26%              | 0.06%    | 0.48% | 1.35%              | 5.37%              |
> > | Dialog | HealthCareMagic | temp T=1        | 3.72%              | 0.02%    | 0.18% | 0.58%              | 2.94%              |
> > | Dialog | HealthCareMagic | Greedy          | 8.27%              | 0.02%    | 1.41% | 1.75%              | 5.09%              |
> >
> > | Task      | Dataset | Decoding Method | Memorization Ratio | Verbatim | Idea  | Paraphrase (p>0.5) | Paraphrase (p<0.5) |
> > |-----------|---------|------------------|---------------------|----------|-------|---------------------|--------------------|
> > | Sentiment | IMDB    | Top-K K=40       | 1.02%              | 0.01%    | 0.13% | 0.18%              | 0.70%              |
> > | Sentiment | IMDB    | Top-p 0.8<p<1    | 1.08%              | 0.01%    | 0.12% | 0.22%              | 0.73%              |
> > | Sentiment | IMDB    | temp T=1         | 0.89%              | 0.01%    | 0.07% | 0.19%              | 0.62%              |
> > | Sentiment | IMDB    | Greedy           | 0.80%              | 0.04%    | 0.30% | 0.17%              | 0.29%              |
> >
> > > **(c) How much do hyperparameters during fine-tuning affect the memorization results? Does a lower learning rate and longer training time result in more memorization?**
> >
> > **Response:** Thank you for your insightful queries. Our empirical observations reveal distinct patterns in memorization across tasks. Specifically, in low-memorization tasks such as sentiment classification, the memorization ratio remains consistently low, irrespective of the model's training adequacy. Conversely, in high-memorization tasks like dialogue, adequate training is crucial, as an undertrained model tends to exhibit lower memorization.
> >
> > To ensure the robustness of our experiments, we calibrated the performance of our fine-tuned models to align with that of Flan-T5. Given that Flan-T5 is a well-trained model demonstrating strong performance across a spectrum of tasks, using it as a benchmark allowed us to ascertain the comprehensive training of our models. This approach ensures that our findings are reflective of well-trained model behavior in various memorization contexts.

---

> ### Author Response · Authors · 2023-11-20
> **A friendly reminder**
>
> We are grateful for the useful comments provided by you. We hope that our answers have addressed your concerns. If you have any further concerns, please let us know. We are looking forward to hearing from you.

---

> > ### Comment · Reviewer_gHsj · 2023-11-21
> > **Response to the authors**
> >
> > Thank you for such a detailed response.
> >
> > A fundamental question that can arise is "Why is it a bad thing for the model to memorize for summarization and dialog tasks? How can you quantify the difference between summarization and memorization? E.g. in your example, how can you quantify if the model memorizes or summarizes a given input?".
> >
> > I had asked about memorization definition before because there were numbers (still present e.g. in section 3.2) like 207% and 196% in the paper. I think they are typographical issues.
> >
> > I believe the paper is interesting. Anyway, I maintain my score (for now) and will discuss it with my fellow reviewers during the discussion period.

---

> > > ### Author Response · Authors · 2023-11-21
> > > **Reply to Reviewer gHsj**
> > >
> > > Thank you for your valuable reply and follow-up questions. We appreciate your positive feedback on this work. Below we provide a detailed explanation of your question.
> > >
> > > > "Why is it a bad thing for the model to memorize for summarization and dialog tasks?
> > >
> > >
> > > **Response:** Thank you for your question. We first want to clarify that in this work, we check how the finefuned LM memorizes the **training data** rather than the test data. There are many applications where we don't want the model to memorize and output **training data** during inference, especially when the training dataset is private or copyrighted. For instance, developing a **diagnostic chatbot**  requires collecting **sensitive medical conversation data.** Similarly, an **academic LM**  may be trained on **copyrighted passages** for summarization or paraphrase generation.
> > > Leakage of such training data can seriously violate user privacy or intellectual property rights:
> > > * Dialog: e.g. If we type "Hello" to the conversational chatbot, the chatbot outputs "I am 20 years old, height 51, height 51, weight 40 kg. I am 6 months pregnant and I can't stop vomiting" (this is a  piece training data to train the model). This would cause a huge privacy invasion for data providers.
> > > * Summarization: Memorization increases the risk of plagiarism, as the model might reproduce text from its training data verbatim or with minimal alterations. This is a significant concern in academic, professional, and creative contexts where originality and intellectual property are highly valued.
> > >
> > >
> > >
> > > > How can you quantify the difference between summarization and memorization? E.g. in your example, how can you quantify if the model memorizes or summarizes a given input?".
> > >
> > > **Response:** Since we only feed the prefix (the first few tokens) into the model, it is easy to distinguish between summarization and memorization. Summarize means that the model summarizes x_prefix(sum(x_prefix)) but does not output the remaining x_suffix (or sum(suffix)). However, for the sake of memory, the model generates information about x_suffix. Therefore, in our experiments we only compare the generated text f(prefix) with x_suffix rather than x_prefix (or the whole x) to see if they are similar.
> > >
> > >
> > > ```
> > > e.g.
> > > X: I am not comfortable from the begining of the month,I am 20 years old, height 51,height 51, weight 40kg.I have been pregnant for 6 months and I can not stop Vomit.
> > >
> > > Input:  I am not comfortable from the begining of the month,I am 20 years
> > > Output:  20 years old, height 5 1 & weight 40kg(Memorization)
> > > Output: 20 years old woman feels bad for a month.(Summarization)
> > >
> > > We only compare output with "height 51,height 51, weight 40kg.I have been pregnant for 6 months and I can not stop Vomit." to identify memorization.
> > > ```
> > >
> > > > I had asked about memorization definition before because there were numbers (still present e.g. in section 3.2) like 207% and 196% in the paper. I think they are typographical issues
> > >
> > > Thanks for pointing this out. In the original paper we use 207\textperthousand. We have modified the format to 20.7% based on your suggestion.
> > >
> > > We hope that our answers have addressed your concerns and merit a better score, if you have any further concerns, please let us know.